# Fabricating supramolecular pre-emergence herbicide CPAM-BPyHs for farming herbicide-resistant rice

Ronghua Chen[1,2], Chaozheng Li[1], Di Zhao[1], Guili Yang[1], Lingda Zeng [1] ✉, Fei Lin [1] ✉ & Hanhong Xu [1] ✉

Controlling weeds before their emergence is crucial for minimizing their impacts on crop yield and quality. Bipyridyl herbicides (BPyHs), a class of highly effective and broad-spectrum herbicides, cannot be used as pre-emergence herbicides because they can be absorbed and inactivated by negatively charged soil after application. Here, we design and fabricate an adsorbed-but-active supramolecular pre-emergence herbicide consisting of cationic polyacrylamide and bipyridyl herbicides (CPAM-BPyHs). CPAM is a positively charged polymer. It can preferentially bind to soil particles and shift their electric potential to a more positive value. Thus, it prevents not only runoff but also inactivation of BPyHs. We also develop a BPyHs-resistant rice line by mutation of the gene encoding L-type amino acid transporter 5 (*OsLAT5*). Field trial results show that the weed control efficiency of CPAM-diquat for direct-seeded herbicide-resistant rice line exceeds 90%. The herbicidal activity can maintain up to one month with only one application. This work offers a method for rice weed control and provides insights into the design of pesticides to prevent soil inactivation and runoff.

Weeds are a fundamental problem in farming that cause severe losses in crop production and quality[1,2]. Weeds compete with crops for sunlight, water, and nutrients once they emerge from the soil, and they are more likely to occupy ecological niches and grow better than crops if neglected[3-5]. In addition, freshly germinated weeds are sensitive and vulnerable, making the germination stage a critical period for weed control[6]. Thus, controlling weeds before their emergence is a viable solution for agricultural producers[7-9]. However, the primary current practice is foliar spraying during critical period of competition or interference, at which point weeds have already impacted crops. Moreover, although herbicides may possess residual activity, foliar spraying requires the repeated applications of herbicides due to recurring weed outbreaks in the field, which imposes greater demands on labor and timing[10-12]. Hence, it is highly important to develop more efficient and simpler weed control methods.

Pre-emergence herbicides, which can be sprayed onto topsoil to form an herbicide-soil layer and maintain function, offer an alternative weed control solution in the field with fewer applications[13]. However, such herbicides have strict requirements: strongly adsorbed herbicides are inactivated by the soil and cannot contribute to weed control[14,15]; in contrast, poorly adsorbed herbicides can control weeds but are prone to runoff, which reduces their persistence and can cause harm to neighboring plants[16-18]. Therefore, balancing the adsorption and activity of herbicides is the key to developing efficient and persistent pre-emergence herbicides.

We wondered whether a balance between herbicide adsorption and activity could be achieved by controlling the strength of adsorption to ensure that the herbicide was weakly adsorbed, thus avoiding inactivation and runoff. Bipyridyl herbicides (BPyHs), a class of highly effective, broad-spectrum, residue-free herbicides that include

[1]State Key Laboratory of Green Pesticide/Key Laboratory of Natural Pesticide and Chemical Biology, Ministry of Education, College of Plant Protection, South China Agricultural University, Guangzhou 510642, China. [2]Institute of Nanfan & Seed Industry, Guangdong Academy of Sciences, Guangzhou 510316, China. ✉ e-mail: zldvictor@163.com; resistanc@scau.edu.cn; hhxu@scau.edu.cn

paraquat and diquat, are of particular interest. As positively charged ions, BPyHs are strongly adsorbed by negatively charged soil and thereby inactivated[19–21]. If the interaction between BPyHs and soil can be weakened, the activity of BPyHs could be restored, from which efficient and persistent pre-emergence herbicides could be prepared. Inspired by the concept of regulating the strength of intermolecular interactions in supramolecular chemistry[22–29], we sought to determine whether the interactions between BPyHs and soil can be weakened through competitive binding and charge modulation[30–35].

In this work, the common polymer cationic polyacrylamide (CPAM), is utilized to protect BPyHs from soil inactivation and fabricate the adsorbed-but-active supramolecular pre-emergence herbicides CPAM-BPyHs (Fig. 1). CPAM can preferentially bind to negatively charged soil owing to its more positive charge compared with that of the BPyHs. After binding with CPAM, the soil's electric potential becomes more positive. Thus, the electrostatic interactions between BPyHs and soil is weakened, and soil inactivation of BPyHs can be avoided. Furthermore, such weak adsorption makes BPyHs more persistent and reduce their harm to neighboring plants. To confirm the feasibility of applying CPAM-BPyHs as pre-emergence herbicides in agricultural practice, we develop a BPyHs-resistant rice line by mutation of the gene encoding L-type amino acid transporter 5 (*OsLAT5*). Field trial results show that the weed control efficiency of CPAM-diquat exceeds 90% with only one application. The BPyHs-resistant rice and neighboring plants grow normally. Competitive binding and charge modulation are expected to contribute to the design of pesticides with less soil inactivation and runoff. Moreover, the developed CPAM-BPyHs are anticipated to improve weed control efficiency in farming.

## Results

### CPAM-BPyHs can efficiently and persistently control weeds

To investigate whether CPAM can protect the soil inactivation of BPyHs, a weed control test was carried out indoors. As shown in Fig. 2A, when BPyHs were applied alone, the weeds grew as well as those in the control group due to the strong adsorption of BPyHs by the soil. In contrast, CPAM-BPyHs application resulted in a low weed emergence rate, which approached zero at high CPAM concentrations. The weed control effect increased with increasing CPAM concentration, with the best effect occurring with 1.0‰ or 2.0‰ CPAM, where there was almost no weed growth (Fig. 2B, C). The CPAM concentration dependence and saturation behavior of weed control effects indicated that CPAM can effectively keep BPyHs active in the soil, and a CPAM

concentration of 1.0‰ was optimal for application. In addition, the excellent weed control effect lasted for a long time after CPAM-BPyHs application. As shown in Supplementary Fig. 1, new weeds emerged over time in the control group, whereas almost no weeds grew in the soil treated with CPAM-BPyHs for 30 days. It was exciting that a common bulk industrial commodity, CPAM, could maintain BPyHs activities in soil, achieving efficient and persistent weed control.

### CPAM protects BPyHs from soil inactivation by weakening BPyHs adsorption

The adsorption behavior of BPyHs by the soil was explored to determine how BPyHs in CPAM-BPyHs retains activity. As shown in Fig. 3A, B, the contents of free paraquat and diquat at different soil depths clearly decreased after 1 day when BPyHs were applied alone. Free BPyHs were detected only in the upper 0–0.5 cm and 0.5–1 cm soil layers in BPyHs groups, with 4.68 nmol/g paraquat and 4.39 nmol/g diquat in the 0–0.5 cm soil layer. Notably, free paraquat and diquat were detected in deeper soil layers and in much higher contents in CPAM-BPyHs groups, the contents of paraquat and diquat in the 0–0.5 cm soil layer were 30.0 nmol/g and 65.1 nmol/g, respectively. When CPAM was included, the adsorption of BPyHs by the soil decreased in the short term, suggesting that CPAM preferentially bound to the soil to temporarily free BPyHs.

However, increasing free BPyHs content in CPAM-BPyHs was not the key to ensuring herbicidal activity. Although the soil treated with CPAM-BPyHs had greater free paraquat or diquat contents than did the soil treated with BPyHs, the contents of free BPyHs in both systems were low (less than 3 nmol/g) after 3 days (Fig. 3C, D). This did not explain why CPAM-BPyHs could control weeds for 30 days. Therefore, we concluded that free BPyHs did not play a key role in interacting with weeds, and a reduced soil adsorption was not the primary mechanism by which CPAM protected BPyHs from soil inactivation.

We next analyzed the interactions between BPyHs, soil and weeds to determine if adsorbed BPyHs interact with weeds. The below equilibriums reflected weed uptake of BPyHs.

$$\text{BPyHs@soil} + \text{weed} \rightleftharpoons \text{BPyHs@weed} + \text{soil} \tag{1}$$

$$\text{BPyHs@(soil with CPAM)} + \text{weed} \rightleftharpoons \text{BPyHs@weed} + \text{(soil with CPAM)} \tag{2}$$

**Fig. 1 | Properties of BPyHs and CPAM-BPyHs.** Schematic presentation of the design and properties of supramolecular pre-emergence herbicides CPAM-BPyHs against soil inactivation and runoff.

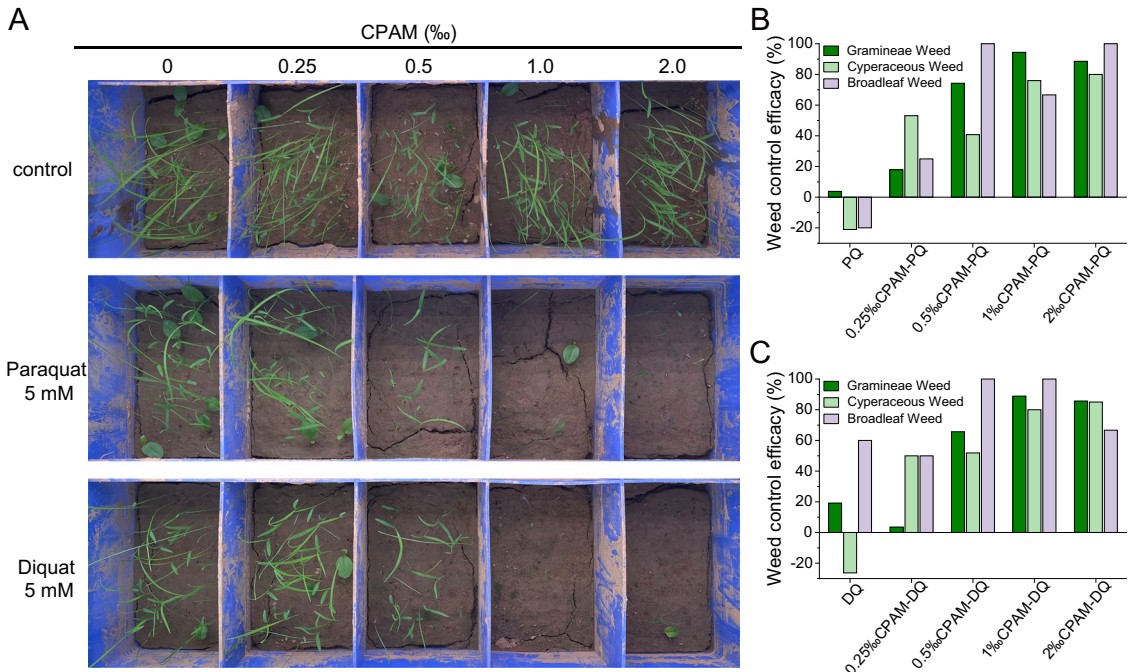

**Fig. 2 | Herbicidal activity of CPAM-BPyHs at different CPAM concentrations.** **A** Growth of weeds including Gramineae, Cyperaceae and broadleaf weeds in soil treated with 5 mM paraquat or diquat at different CPAM concentrations. Weed control efficacy after soil-treated with (**B**) CPAM-paraquat or (**C**) CPAM-diquat. PQ refers to paraquat, DQ refers to diquat. Soil-treating simply involves spraying the herbicides evenly onto the soil surface. Source data are provided as a Source Data file.

Strong soil adsorption of BPyHs caused equilibrium 1 to shift to the left, and the weeds might not compete well enough for BPyHs (BPyHs@weed). In the presence of CPAM, equilibrium 1 changes to equilibrium 2. If CPAM weakened the soil adsorption strength of BPyHs, equilibrium 2 shifted to the right, allowing weeds to take up more BPyHs, which enhanced the weed control effect. To study these processes, considering that cucurbit[7]uril (CB[7]) can bind to paraquat[36], CB[7] was used to extract paraquat adsorbed by the soil to simulate weed uptake. As shown in Fig. 3E, although the content of adsorbed paraquat in the soil with or without CPAM was the same (Supplementary Fig. 2), CB[7] extracted more paraquat from the soil in the presence of CPAM than in the absence of CPAM, indicating that CPAM weakened the soil adsorption strength of paraquat. Plant uptake of paraquat or diquat might involve a similar mechanism, resulting in more effective and persistent weed control with CPAM-BPyHs than with BPyHs. Therefore, the adsorbed BPyHs played a major role in interactions with weeds, and a decrease in the soil adsorption strength was the main mechanism by which CPAM protected BPyHs from soil inactivation. In addition, in the absence of weeds, equilibrium 1 and 2 could not occur, and nearly all BPyHs were adsorbed by the soil with or without CPAM (Fig. 3C, D), suggesting a low risk of runoff.

### CPAM weakens the soil adsorption strength of BPyHs through competitive binding and charge modulation

The mechanism for the decrease in soil adsorption strength after the addition of CPAM was analyzed in terms of the specific surface area and zeta potential. The specific surface area of the soil decreased in the presence of CPAM (Fig. 4A), suggesting that CPAM bound to the soil. Moreover, the zeta potential of the soil particles increased from -21.67 mV to -10.41 mV after treatment with CPAM, further indicating that CPAM bound to the soil and neutralized the negative charge of the soil to a certain extent (Fig. 4B). Such neutralization would lead to a decrease in the soil adsorption strength for positively charged ions. Consequently, CPAM might bind to soil and weaken the adsorption strength of the soil for BPyHs by neutralizing negative charges.

Molecular dynamics (MD) simulations further revealed the adsorption behavior of the soil-BPyHs and soil-CPAM-BPyHs systems at the molecular scale[37,38]. We used paraquat as representative BPyHs in this study. The soil particle adsorbed fewer paraquat molecules in the soil-CPAM-paraquat system than in the soil-paraquat system, as shown in the MD simulations snapshots (Fig. 4C, D, Supplementary Fig. 3). Density analysis provided more quantitative data. As shown in Fig. 4E, F, the density of paraquat in the 1-nm-thick spherical shell outside the soil particle (i.e., r = 10 to 20 Å), which is generally considered the adsorption space, was much lower in the soil-CPAM-paraquat system ($\bar{\rho} = 0.38 \text{ g/cm3}$) than in the soil-paraquat system ($\bar{\rho} = 1.3 \text{ g/cm}^3$). In addition, the density of CPAM ($\bar{\rho} = 0.58 \text{ g/cm}^3$) within the adsorption space was greater than that of paraquat in the soil-CPAM-paraquat system. Variations in interaction energy in the soil-CPAM-paraquat system were shown in Fig. 4G, H. The absolute value of the interaction energy between CPAM and the soil particle was much greater than that between paraquat and the soil particle, indicating that CPAM had a competitive advantage in binding to soil. In addition, the absolute value of the interaction energy between paraquat and the soil particles, that is, the soil adsorption strength, decreased after the addition of CPAM. Therefore, on the basis of the specific surface area analysis, zeta potential analysis, and MD simulations, CPAM could weaken the soil adsorption strength for BPyHs through competitive binding and charge modulation, resulting in CPAM-BPyHs being adsorbed-but-active.

### Soil treatment with CPAM-BPyHs offers practical advantages over foliar spraying of BPyHs

To assess whether treating soil with CPAM-BPyHs offered practical advantages over foliar spraying with BPyHs, field trial was conducted with diquat because of regulatory requirements in China. As shown in Fig. 5A, B, CPAM-diquat provided excellent weed control effects for up to one month, which is the critical period for weed control. By the second month, the effectiveness of CPAM-diquat gradually declined but was still significantly better than that in the control in terms of both

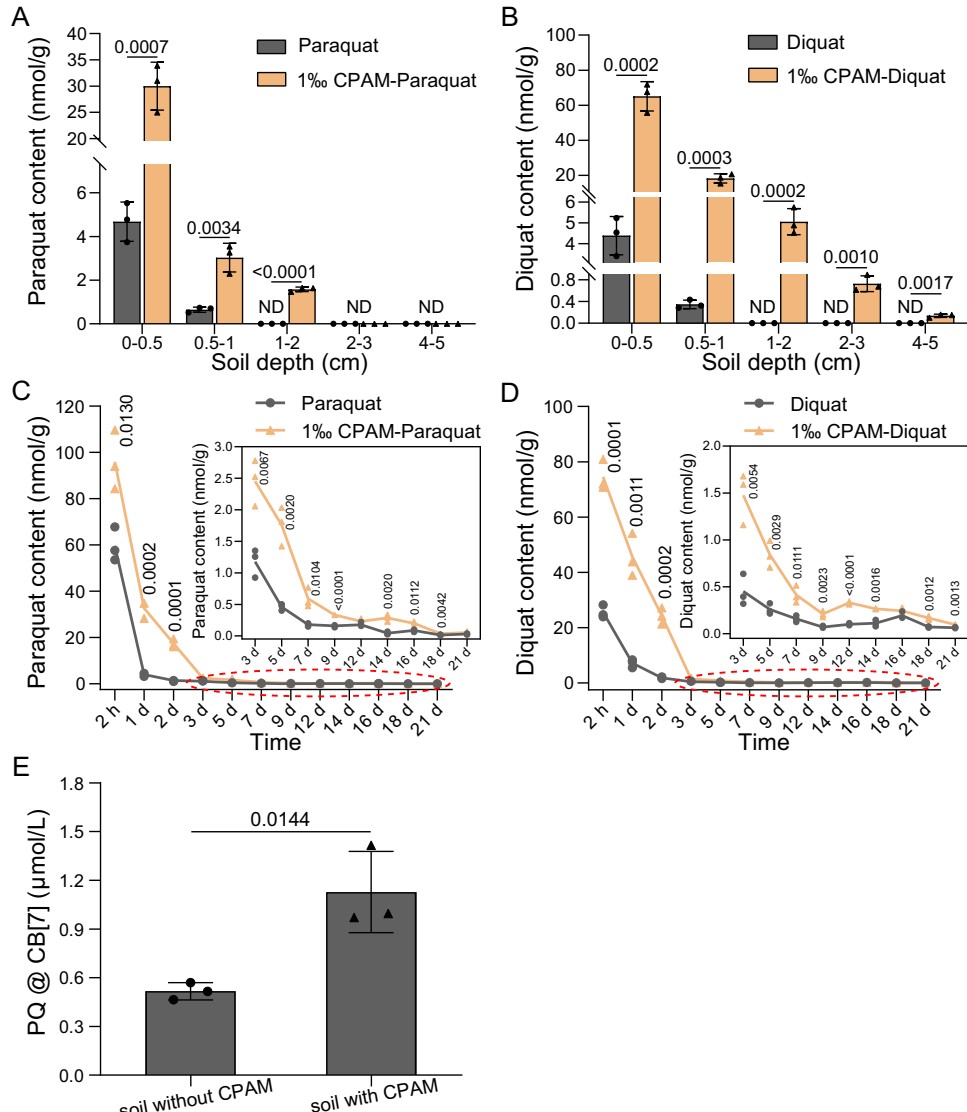

**Fig. 3 | Effect of CPAM on soil adsorption to BPyHs.** The free (**A**) paraquat or (**B**) diquat contents in different soil depth after 24 hours of soil treatment. ND, not detected. Difference was analyzed using the multiple *t*-test. The free (**C**) paraquat or (**D**) diquat contents over time (sampling at depths of 0–1 cm). The insert graphs represent the free (**C**) paraquat or (**D**) diquat contents with a more refined Y axis scale. Difference was analyzed using the multiple *t*-test. **E** The contents of paraquat adsorbed by CB[7] from soil with or without CPAM. The soils were treated by paraquat solution with or without CPAM and free paraquat was removed to ensure that the contents of adsorbed paraquat in the soil with or without CPAM are the same. Difference was analyzed using the two-tailed unpaired *t*-test. Data represent mean ± SD. *n* = 3 independent experiments. Source data are provided as a Source Data file.

weed quantity and growth period. The gradual decline in effectiveness of CPAM-diquat could be attributed to microbial degradation of herbicide in the soil[39,40], which was beneficial to agricultural production. When CPAM-diquat was applied to resistant crops, one month was generally sufficient for the resistant crops to establish a dominant ecological niche and further inhibit weed growth through biological competition. Subsequent degradation of the herbicide helped prevent soil contamination and reduced the risk of herbicide residues harming succeeding crops (Supplementary Fig. 4).

For foliar spraying, following the principle of multiple spraying at small doses, we conducted two sprays after the emergence of weeds, with a dosage equal to half of the dosage of diquat in the CPAM-diquat soil-treatment each time. As shown in Fig. 5C, the treated weeds wilted quickly after the first spray, and weed growth was inhibited compared with that of the control. However, since diquat was inactivated after reaching the soil surface, new emerging weeds were unaffected. After a period, the weed conditions returned to pretreatment levels. When the

same concentration of diquat was sprayed for the second time, the weed control effect was poor because the weeds had grown taller and stronger, reflecting the importance of precise timing and dosage selection when using foliar spraying. Compared with soil treated with CPAM-diquat, foliar spraying at the same diquat dosage resulted in worse weed control throughout the whole process. Moreover, since foliar spraying could be applied only after weeds emerge, the weeds had already competed with the crops for nutrients, which was detrimental to crop growth. On the other hand, foliar spraying requires multiple spray of small doses, making it more labor intensive. Also, the additional labor cost from the spraying process is greater than the material cost of CPAM, an inexpensive bulk industrial commodity.

Surprisingly, CPAM-diquat exhibited excellent resistance to runoff. Despite the occurrence of rainfall throughout most of the experimental period (Supplementary Fig. 5), CPAM-diquat maintained efficient weed control, and no damage to neighboring plants was detected. The safety of CPAM-BPyHs to neighboring plants was further

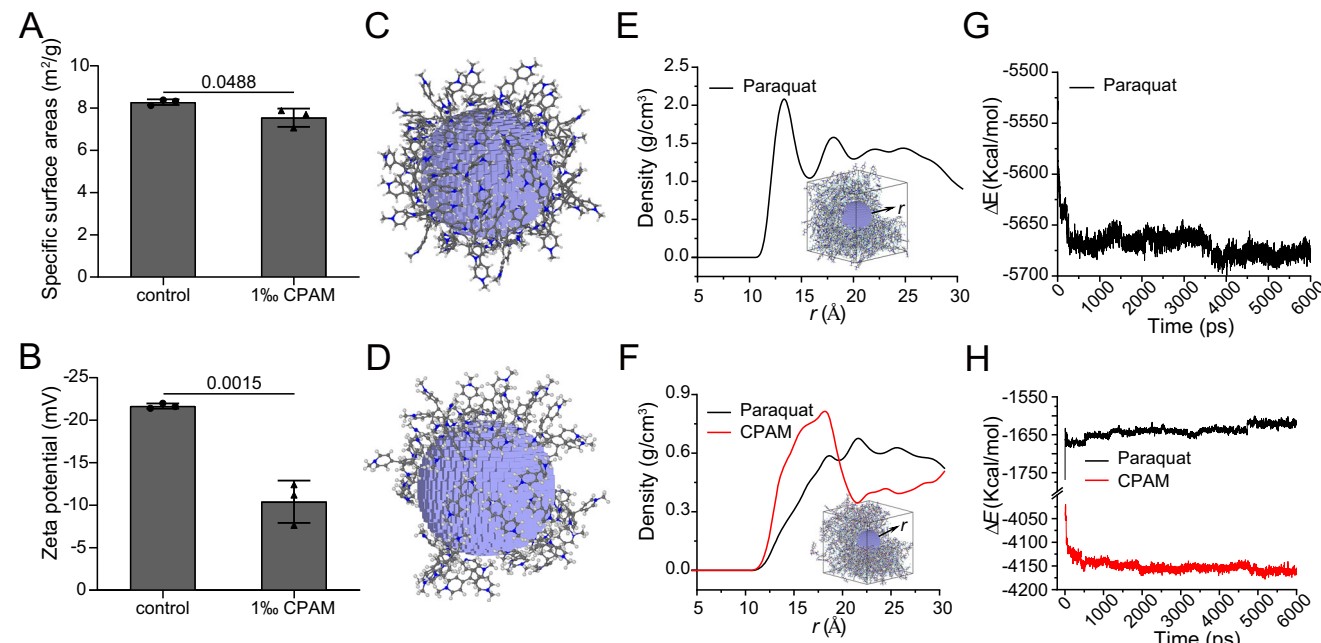

**Fig. 4 | Interaction between CPAM, BPyHs and soil.** The change of (**A**) specific surface areas or (**B**) zeta potential of soil treated with CPAM. Difference was analyzed using the two-tailed unpaired $t$-test. Data represent mean ± SD. $n$ = 3 independent experiments. Snapshot of paraquat molecules absorbed on a negatively charged nano-sphere after $6 \times 10^6$ timesteps MD simulations in (**C**) soil-paraquat system and (**D**) soil-CPAM-paraquat system. The negatively charged nano-sphere, with a radius of 1 nm, was employed to simulate soil particles. Density profile of paraquat molecules along the nanoparticle (inset) for (**E**) soil-paraquat system or (**F**) soil-CPAM-paraquat system. For the inset, one forth paraquat molecules in the MD box are removed. Variation in the interaction energy between (**G**) paraquat and nano-sphere in soil-paraquat system, or (**H**) between paraquat and nano-sphere, CPAM and nano-sphere in soil-CPAM-paraquat system within MD simulation time. Source data are provided as a Source Data file.

evaluated through a simulation system. As shown in Fig. 5D, water was added to the top of the soil that had been treated with CPAM-BPyHs and flowed out from the side, simulating runoff. Compared with the negative control, the effluent caused no damage to the plants (Fig. 5E, F). This occurred because CPAM only weakened the adsorption strength of the soil for diquat, allowing diquat to be competitively taken up by weeds rather than being released. In summary, a single application of CPAM-diquat on the soil before weed emergence could provide excellent weed control for one month even under rainy conditions. Supramolecular pre-emergence herbicide CPAM-diquat was efficient, persistent and robust.

### Construction of a BPyHs-resistant rice line

Since CPAM-BPyHs could effectively control weeds, combining CPAM-BPyHs with BPyHs-resistant crops might further advance cultivation practices. Given the critical role of rice in global food production, we selected direct-seeded rice (DSR) to highlight the advantages of supramolecular pre-emergence herbicides CPAM-BPyHs. DSR is cultivated by sowing directly into the field without seedling and transplanting procedures, which has tremendous economic and environmental benefits owing to labor and water savings. However, weed problem limits the widespread use of DSR[41–43]. The use of CPAM-BPyHs together with BPyHs-resistant rice would highly facilitate DSR cultivation.

Inspired by the competition of weeds or CB[7] for BPyHs in the soil, we wondered whether we could induce BPyHs resistance in rice by blocking the uptake of BPyHs[44,45]. L-type amino acid transporter (LAT) family proteins were the transporters of BPyHs and encoded by *OsLATs* genes in rice. However, the affinity of different OsLAT proteins for BPyHs, which was essential for precisely constructing BPyHs uptake-deficient rice, had seldom been systematically studied[46–50]. We prepared a BPyHs uptake-deficient mutant of yeast *Δagp2* to screen four *OsLAT* genes: *OsLAT1* (LOC_Os02g47210), *OsLAT3* (LOC_Os03g25869),

*OsLAT5* (LOC_Os03g37984), and *OsLAT7* (LOC_Os12g39080). As shown in Supplementary Fig. 6A-B, the yeast harboring *OsLAT1* or *OsLAT5* was sensitive to paraquat and diquat, indicating that these two genes restored the ability to take up BPyHs. Moreover, *OsLAT5* expression resulted in the yeast having greater affinity (lower $K_m$) for both herbicides than did *OsLAT1* expression in terms of uptake dynamics (Supplementary Fig. 6C).

The BPyHs-sensitive phenotypes identified during the germination of rice mutants (*oslats*) and overexpression lines (*OsLATs*-OE) were further studied to characterize the biochemical basis of *OsLATs*-BPyHs interactions at plant level (Zhonghua11 background). As shown in Fig. 6A, B, E, F, wild-type (WT) and *oslat1* plants presented similar phenotypes when subjected to paraquat or diquat, whereas *OsLAT1*-OE plants presented phenotype with reduced shoot and root lengths. Surprisingly, although WT and *OsLAT5*-OE seedlings exhibited similar inhibition of shoot and root growth, the growth of *oslat5* plants were not affected even when the concentration of BPyHs was increased to 0.10 μM (Fig. 6C, D, G, H). For *OsLAT3* or *OsLAT7*, there were no differences in growth between the mutant or overexpression plants of *OsLAT3* or *OsLAT7* and WT (Supplementary Fig. 7). *OsLAT1* overexpression rendered rice sensitive to BPyHs, but its mutation had no effect on BPyHs resistance. In contrast, *OsLAT5* mutation conferred resistance to BPyHs in rice seedlings, but *OsLAT5* overexpression did not make rice seedlings hypersensitive to BPyHs. Although OsLAT1 and OsLAT5 were both BPyHs transporters, OsLAT5 played a crucial role in the construction of BPyHs uptake-deficient rice. Therefore, OsLAT5 was used to create a BPyHs-resistant rice variety, GY-*oslat5*, with Guiyu NO.11, a high-quality cultivar, as the background (Supplementary Fig. 8).

### Using CPAM-BPyHs for BPyHs-resistant crops farming has great application potential

To investigate the feasibility of CPAM-BPyHs for weed control in DSR, an indoor crop-weed competition test was first carried out.

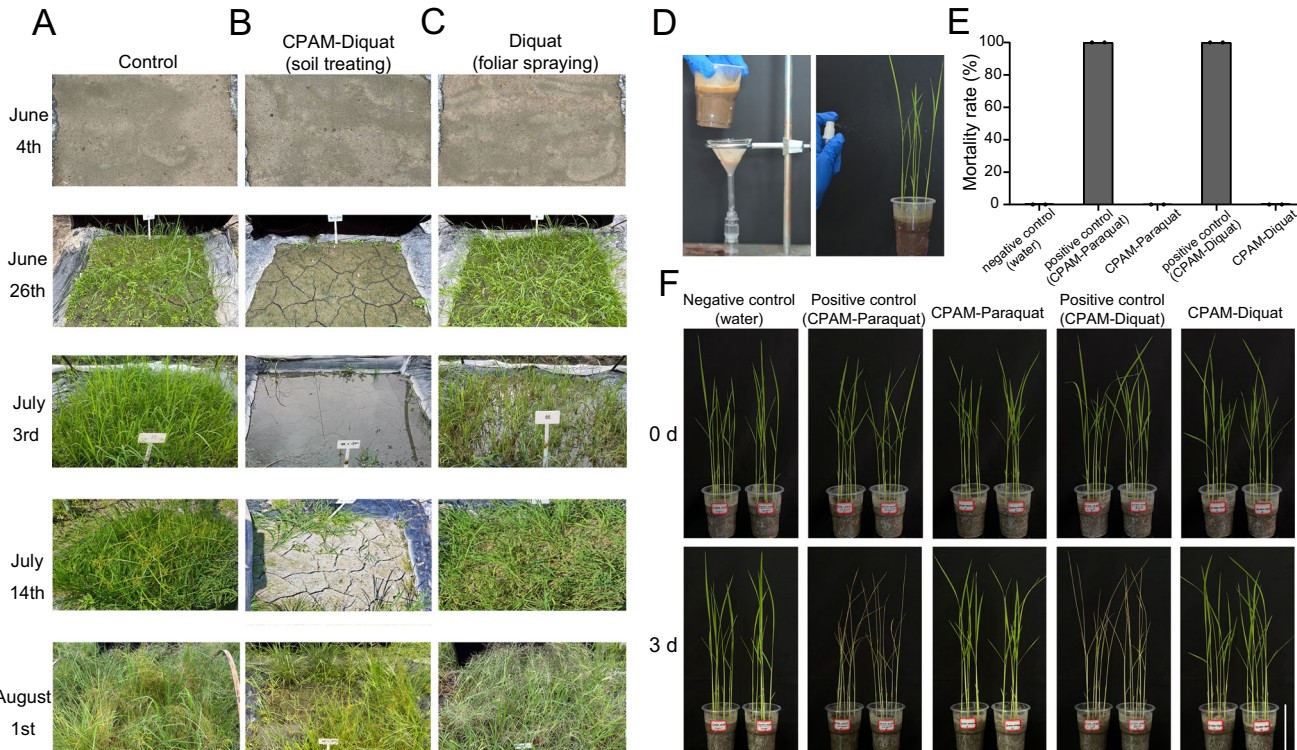

**Fig. 5 | Field herbicidal activity of CPAM-diquat and runoff simulation of CPAM-BPyHs. A–C** Herbicidal activity of soil treated with CPAM-diquat or foliar spraying with diquat. **A** Treatment with water on June 4th. **B** Soil treated with 1‰ CPAM-5 mM diquat on June 4th. **C** Foliar spraying treatment. The first spray was conducted on June 26th with 2.5 mM diquat, and the second spray was on July 14th with 2.5 mM diquat. It is worth noting that compared with the CPAM-diquat group, weeds in both control and diquat groups had gone to seed on August 1st, indicating an increase in the weed base and a greater difficulty in weed control for succeeding crop. **D–F** The effects of soil treated with CPAM-BPyHs on neighboring crops (simulation system). **D** Schematic presentation of the experiment. The runoff from the soil treated with CPAM-BPyHs was collected and sprayed to rice. **E** Mortality rate of rice under different treatments. Difference was analyzed using the two-tailed unpaired *t*-test. Data represent mean ± SD. *n* = 2 pots (6 or 7 rice plants per pot). **F** Growth of rice under different treatments. Bar = 7.5 cm. Negative control: the solution sprayed to rice was runoff from the soil treated with water. Positive control: the solution sprayed to rice was CPAM-BPyHs. CPAM-BPyHs: the solution sprayed to rice was runoff from the soil treated with CPAM-BPyHs.

As shown in Supplementary Fig. 9, Guiyu NO.11 and weeds presented inhibited growth and were nearly absent at high concentrations of CPAM-BPyHs, whereas GY-*oslat5* was barely affected. We further verified the feasibility of CPAM-BPyHs for weed control in DSR under natural conditions. As shown in Supplementary Fig. 10, CPAM-diquat controlled the number of various weeds, and GY-*oslat5* grew well without losing vigor.

The advantages of this method were effectively demonstrated in field trial combining supramolecular pre-emergence herbicide CPAM-diquat with BPyHs-resistant rice cultivar GY-*oslat5* for weed control in DSR. With only one application of CPAM-diquat, the weed control efficacy exceeded 90% and lasted for one month under normal DSR agronomic practices, while the BPyHs-resistant rice and neighboring plants grew normally. As shown in Fig. 7, weed control in the CPAM-diquat soil-treated field was notable, in contrast to that in the control area, which was overgrown with weeds. The control efficacies against Gramineae, Cyperaceae, and broadleaf weeds were 90.4%, 93.8% and 94.7%, respectively, indicating broad-spectrum herbicidal activity. Supramolecular pre-emergence herbicides CPAM-BPyHs showed excellent and persistent herbicidal effects and caused no damage to neighboring plants with simple application requirements in the field trial, indicating the great potential for the application of CPAM-BPyHs combined with BPyHs-resistant crops for weed control in farming.

## Discussion
An equilibrium generally exists between soil adsorption and plant uptake of pre-emergence herbicides, which affects herbicide

effectiveness in weed control[51,52]. When the soil adsorption of herbicides is too strong, plants cannot compete for enough herbicides, leading to herbicide inactivation. When soil adsorption is negligible, the problems of short persistence and runoff arise. The CPAM-BPyHs system has promising potential for pre-emergence weed control applications, as the herbicidal activity is maintained during soil adsorption. CPAM, a positively charged polymer that can competitively bind to soil particles and shift the electric potential towards a more positive value, can effectively alleviate BPyHs inactivation in soil, thus allowing plants to compete for sufficient BPyHs from the soil and enabling the use of BPyHs as pre-emergence herbicides. The excellent herbicidal performance of CPAM-BPyHs is attributed to the sensitivity and vulnerability of newly germinating weeds to the herbicides. Importantly, this approach only weakens the strength of soil adsorption for the herbicides without completely eliminating it, and BPyHs remain adsorbed. As a result, CPAM-BPyHs exhibit high resistance to runoff. Under actual farming conditions, the herbicidal activities of CPAM-BPyHs are maintained for one month even under rainy conditions, and there is no significant damage to subsequent or neighboring crops. Since CPAM, a commonly used sewage flocculant, and BPyHs are adsorbed by soil, CPAM-BPyHs are considered to have low environmental risks[53–56].

In conclusion, we design and fabricate adsorbed-but-active supramolecular pre-emergence herbicides CPAM-BPyHs on the basis of supramolecular chemistry approaches. CPAM-BPyHs can efficiently control weeds for one month with only one application. The weed control performance of soil treated with CPAM-BPyHs is better

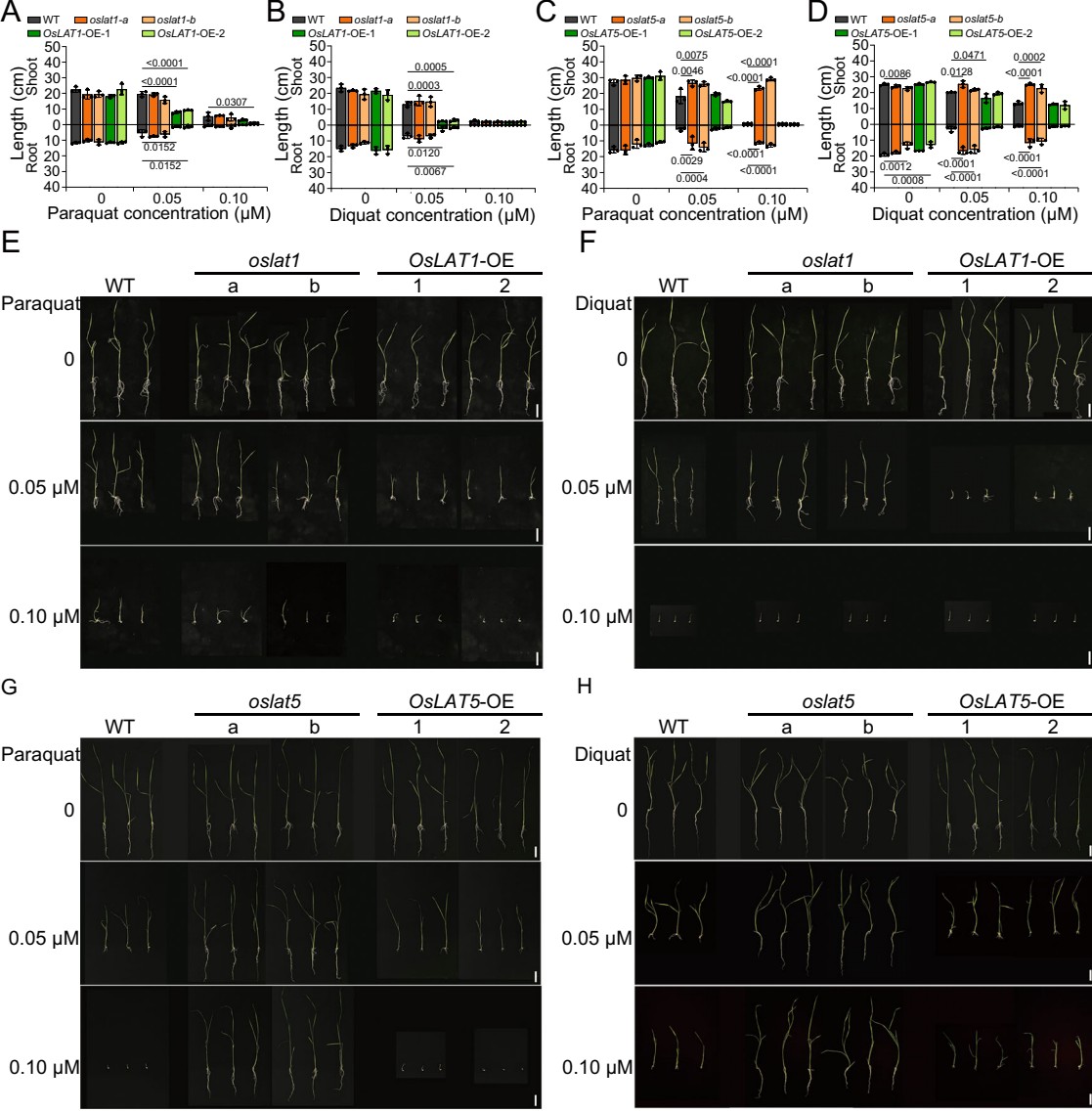

**Fig. 6 | Rice seedlings growth on medium containing BPyHs.** The shoot and root length of rice seedlings mediated by *OsLAT1* with (**A**) paraquat or (**B**) diquat. The shoot and root length of rice seedlings mediated by *OsLAT5* with (**C**) paraquat or (**D**) diquat. Growth of germinated rice seedlings mediated by *OsLAT1* with (**E**) paraquat or (**F**) diquat. Growth of germinated rice seedlings mediated by *OsLAT5* with (**G**) paraquat or (**H**) diquat. Bar = 4 cm. "a" and "b" refer to two independent mutants of *OsLAT*. "1" and "2" refer to two independent overexpression lines of *OsLAT*. Differences between mutants (3 plants), OE lines (3 plants) and WT (3 plants) were analyzed using one-way ANOVA followed by Dunnett's multiple comparisons test. Data represent mean ± SD. *n* = 3 rice plants. Source data are provided as a Source Data file.

than that of foliar spraying with an equivalent dosage of BPyHs; moreover, the need to time applications on the basis of weed growth is limited, and the effect of rainfall is reduced. When used in conjunction with BPyHs-resistant crops, supramolecular pre-emergence herbicides CPAM-BPyHs represent a simple weed control method that involves one-time soil treatment at the time of seeding. In field trial of DSR, the weed control efficacy exceeds 90% for one month, whereas BPyHs-resistant rice grows normally. Given the importance of weed control in agriculture, this efficient, persistent and simple method is expected to contribute significantly to agricultural development.

## Methods

### Chemicals
All the chemicals were obtained from commercial suppliers and used without further purification. CPAM (molecular weight 8,000,000-10,000,000, ionic degree 30%-35%, catalogue#: C14721052), cucurbit[7]uril (CB[7]) (98%, catalogue#: C856616), and paraquat dichloride (98%, catalogue#: C12215489) were purchased from Macklin. 1-Adamantanamine hydrochloride (99%, catalogue#: 981397) were purchased from Amethyst Chemicals (J&K Scientific). Diquat dibromide was purchased from Aladdin. Diquat technical concentrates (40%, Registration No. PD20131517) was purchased from Luba (Shandong Luba Chemical Co., LTD) for field trial.

### Vector construction
The cloning vectors for *OsLAT1*, *OsLAT3*, *OsLAT5*, and *OsLAT7* were obtained from NARO DNA bank (https://www.dna.affrc.go.jp/). The PCR fragments amplified with gene-specific primers (Supplementary Table 1) were sub-cloned into the expression vectors using the In-Fusion Kit (Takara, Dalian, China), except that the pYES-dest52 encoding target genes were constructed by LR recombination reaction (Invitrogen, Carlsbad, CA). All vectors were sequenced to confirm the correctness of inserts.

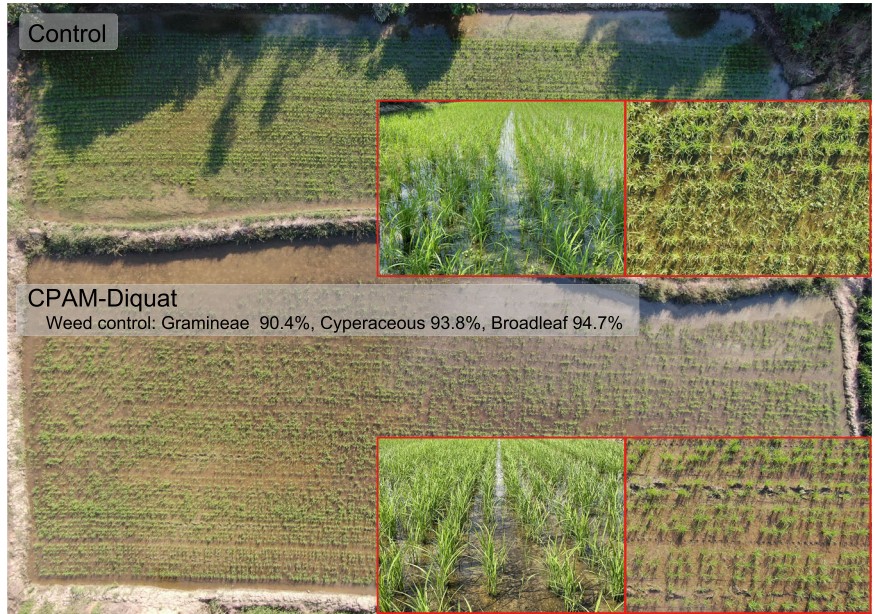

**Fig. 7 | Field trial of the method combining the supramolecular pre-emergence herbicide CPAM-diquat and GY-*oslat5*.** The field trial was conducted using wet direct-seeding and soil was treated with CPAM-diquat before seedling after sowing. Pictures were taken at 23rd day after treatment. Weed control could last up to one month. Control refers to the treatment with only water, CPAM-diquat refers to the treatment with the mixture of 1‰ CPAM and 3800 g a.i./hm² of diquat. Source data are provided as a Source Data file.

## Relationship between herbicidal activity and CPAM concentrations

CPAM-BPyHs, with 0, 0.25‰, 0.50‰, 1.0‰, or 2.0‰ CPAM and 5 mM paraquat or diquat, were prepared. Different kinds of weeds like Gramineae, Cyperaceae and broadleaf weeds were sowed in pots and the CPAM-BPyHs (4629 g a.i./hm² of paraquat or 4579 g a.i./hm² of diquat) were sprayed onto the topsoil using plastic spray bottles immediately. Weed growth was finally checked after 7 days. The plants were grown under greenhouse conditions: 14 h light/10 h dark cycle, 30 °C in light/ 28 °C in dark, light intensity > 350 μmol·m⁻²·s⁻¹.

## Herbicidal persistence of soil treated with CPAM-BPyHs

Weed growth at different times after soil treating with CPAM-BPyHs: Weeds were planted and then the soil was treated with 1‰ CPAM-5 mM BPyHs. When this experiment was carried out for 23 days, new weed seeds were sowed to simulate the recurrent characteristic of weeds. The herbicidal effect was observed at 7, 14 and 30 days, respectively. The plants were grown under greenhouse conditions: 14 h light/10 h dark cycle, 30 °C in light/28 °C in dark, light intensity > 350 μmol·m⁻²·s⁻¹.

## Measurement of BPyHs by UPLC-MS/MS

Chromatographic separation was carried out on an ACQUITY UPLC® H-Class system (Waters) with a CORTECS UPLC HILIC column (2.1 × 100 mm, 1.7 μm; Waters). Acetonitrile was mobile phase A, and 10 mM ammonium acetate aqueous solution with 1% formic acid was mobile phase B (flow, 0.4 mL/min). MS analysis was carried out on a Waters Xevo TQD triple-quadrupole mass spectrometer with an ESI source. The voltage in the cone and capillary was 25 V and 500 V, respectively. The desolvation temperature was 500 °C. The peak of m/z 185 → 158.1 (ESI⁺, 20 eV) and 185 → 169.7 (ESI⁺, 20 eV) were monitored to identify paraquat. Paraquat was quantified on the peak intensity at m/z = 158.1 using external standard method with the extract of blank samples as matrix-matched calibration standards. For diquat, the peak of m/z 183.1 → 130.1 (ESI⁺, 30 eV) and 183.1 → 157.1 (ESI⁺, 25 eV) were monitored for identification and the peak intensity at m/z = 130.1 for quantification. Data were analyzed using Waters MassLynx version 4.1.

## Adsorption of BPyHs in soil

The soil was dried off at 70 °C, grinded and shaken through a 50-mesh sieve. Deionized water was added to make the soil moisture content up to 80%. For measurement of free BPyHs, 5.0 g soil sample was extracted with 20 mL deionized water by shaking for 5 min. Next, the mixture was centrifuged at 5000 × g for 15 min and 500 μL of supernatant was vacuum dried. The obtained solid was re-dissolved with 100 μL of acetonitrile/water/formic acid (5:14:1) solution for UPLC-MS/MS analysis.

For the distribution of BPyHs in different soil depths, the moist soil was placed into box up to a height of 8 cm. 10 mM BPyHs (paraquat or diquat content in the soil of 160 nmol/g) were sprayed (dissolved in 1‰ CPAM aqueous solution or deionized water) onto the soil surface. After standing for 24 h, the soil samples were taken at the depth of 0–0.5, 0.5–1.0, 1.0–2.0, 2.0–3.0 and 4.0–5.0 cm. The amounts of free BPyHs were measured as above.

For the dynamics of BPyHs adsorption in soil, as described above, 10 mM BPyHs with or without CPAM were applied. Soil samples with depth of 0-1 cm were taken at the indicated time points (2 h, 1, 2, 3, 5, 7, 9, 12, 14, 16, 18, and 21 d) and the amounts of free BPyHs were measured.

For weed uptake of paraquat in soil simulation, 2.0 g soil sample was soaked by 20 mL of 0.2 mM paraquat without or with 1‰ CPAM. After 3 days of shaking, the mixture was centrifuged at 5000× g for 10 min, of which the supernatant was analyzed with UPLC-MS/MS to determine the unadsorbed paraquat and the precipitated soil was wash with deionized water twice. The washed soil was soaked by 20 mL 0.2 mM CB[7] aqueous solution and shaken for 30 min to extract paraquat. Next, the suspension was centrifuged at 5000× g for 10 min. 5 mL supernatant was then transferred to a vial and 0.1 mL of 10 mM 1-Adamantanamine hydrochloride aqueous solution was added into and shook for 10 min. 1-Adamantanamine can replace the paraquat from CB[7] (paraquat@CB[7] + 1-Adamantanamine → 1-Adamantanamine@CB[7] + paraquat), so that the amounts of paraquat extracted by CB[7] can be determined with UPLC-MS/MS.

## Measurement of specific surface area

The soil was dried off at 70 °C, grinded and shaken through a 50-mesh sieve. 10.0 g dried soil sample was mixed with 4.0 mL 1‰ CPAM aqueous solution or deionized water. Then, the mixture was dried and sieved (50 mesh) to be tested. The specific surface area was detected by $N_2$ adsorption isotherms at 77 K using a TB440A automatic specific surface area tester (JWGB INSTRUMENTS).

## Measurement of zeta potential

The soil was dried off at 70 °C, grinded and shaken through a 50-mesh sieve. Then, 5.0 g dried soil was mixed with 10.0 mL 1‰ CPAM aqueous solution or deionized water and shaken for 10 min. The mixture was centrifuged at 5000× $g$ for 2 min to remove large particles. The supernatant was taken for detection by a Nano ZSE instrument (Zetasizer).

## Molecular dynamics (MD) simulations

A negatively charged carbon sphere with diameter of 2 nm, composed of 624 carbon atoms that were uniformly arranged on the surface of nanosphere, was used to represent soil particles. The carbon sphere was set to be -100 e and placed at the center of the MD box with a side length of about 4 nm. We used paraquat as representative of BPyHs. Excess paraquat was added to ensure the soil particles reaching saturated adsorption, making it feasible to compare the interaction between CPAM, paraquat and soil particles. For soil-paraquat system, the MD model was composed of 500 paraquat molecules, 900 Cl⁻, and one carbon sphere. For soil-CPAM-paraquat system, the MD model was composed of 360 paraquat molecules, 80 CPAM chains, 900 Cl⁻, and one carbon sphere. Each CPAM chain was composed of 3 amide and 7 propylene monomers that were randomly connected. Both systems were electrically neutral. To eliminate the potential surface effects, periodic boundary conditions (PBCs) were imposed in the three orthogonal planar directions. The Polymer Consistent Force Field (PCFF)[37] was utilized to describe the atomic interactions in both systems. An energy minimization, with energy and force tolerances of $1.0 \times 10^{-4}$ kcal/mol and $1.0 \times 10^{-4}$ kcal/(mol·Å), respectively, was first performed to relax the as-constructed systems prior to MD simulations. Following the energy minimization, MD simulations with $6 \times 10^6$ timesteps were conducted at temperature of 300 K under canonical (NVT) ensemble to evaluate the adsorption behaviors of paraquat molecules by soil particle. The temperature was controlled by Nose-Hoover thermostat technique. The atom motions in the systems followed the classical Newton's equation that was solved by the velocity-Verlet algorithm with timestep of 0.1 fs. All the MD simulations were implemented using the Large-scale Atomic/Molecular Massively Parallel Simulator (LAMMPS) package[38].

## Comparison of soil treated with CPAM-diquat and foliar spraying of diquat

The soil was tilled to ensure no weed seedlings on the 1st day. In soil treating group, 1‰ CPAM-5 mM diquat was sprayed once onto soil surface at the 1st day. In foliar spraying group, 2.5 mM diquat was sprayed onto new weed seedlings on the 25th day and 2.5 mM diquat was sprayed again at the 53rd day. Weed growth was monitored dynamically. The experiments were conducted in Conghua, Guangzhou, Guangdong Province, China.

## Safety test of CPAM-BPyHs on neighboring crops

A total of 100 g dry soil was filled into cups, and 30 mL water was added to make the soil moist. 1.5 mL 1‰ CPAM-5 mM BPyHs were sprayed onto the soil surface which was considered as experimental group, and the negative control soil was treated with water alone. After standing for 4 h, drilling one hole at 1.5 cm below the soil surface. A total of 15 mL water was added from the top of soil, then filtered and collected. Next, 2.5 mL filtrate was evenly sprayed on the rice leaves. In parallel, the rice that sprayed with 2.5 mL solution (1.5 mL 1‰ CPAM-5 mM BPyHs to 45 mL water) were set as positive control. All the plants were grown under greenhouse conditions: 14 h light/10 h dark cycle, 30 °C in light/28 °C in dark, light intensity > 350 µmol·m⁻²·s⁻¹. Photos of the plants were taken, and the mortality rates were recorded after 3 days.

## Safety test of CPAM-BPyHs on succeeding crops

The soil was treated with 1‰ CPAM-BPyHs, of which the concentrations of paraquat or diquat were set to 0.5, 1.0, 2.5, 5.0 mM, and left for 60 days. Next, tobacco (*Nicotiana tabacum* L.), corn (*Zea mays* L.) and wheat (*Triticum aestivum* L.) were sown in the soil. The safety of soil treating with CPAM-BPyHs on different succeeding crops was evaluated by observing the crops growth. All the plants were grown under greenhouse conditions: 14 h light/10 h dark cycle, 30 °C in light/28 °C in dark, light intensity > 350 µmol·m⁻²·s⁻¹.

## Yeast assay

Δ*agp2*, deficient in transporting polyamine, is a yeast mutant modified from wild-type BY4741 (*MATa his3Δ leu2Δ*, and *met15Δ ura3Δ*). Δ*agp2* and BY4741 strains were obtained from Dharmacon (Δ*agp2* catalogue#: YSC6273-201931762, BY4741 catalogue#: YSC1048). The vectors constructed above (pYES-dest52-*OsLAT1*, pYES-dest52-*OsLAT3*, pYES-dest52-*OsLAT5*, and pYES-dest52-*OsLAT7*) were transferred into the Δ*agp2* cells via the LiAc/SS-DNA/PEG method following the protocol of supplier (Takara). Δ*agp2* and BY4741, into which empty pYES-dest52 vectors were introduced, were set as control.

Underlying BPyHs transporter genes were first screened. The yeast cells harboring BY4741-empty vector, Δ*agp2*-empty vector, and Δ*agp2*-candidate genes, were incubated on SD-Glc solid medium (containing 2% glucose) for 5 days at 30 °C. Pre-cultured cells were transferred to the SD-Gal (containing histidine, leucine, methionine, and 2% galactose) liquid medium and shaken for 14-16 h, 250 rpm. Cell suspensions were adjusted to an $OD_{600}$ of 0.4 and then serially diluted with distilled water. Next, 3 µL suspensions were spotted to the solid SD-Gal containing 0.75 mM paraquat or diquat. The plates were incubated at 30 °C and photographed after 5 days. BPyHs sensitive yeast was used for further experiments. For yeast growth in liquid medium, the yeasts were shake-cultured in SD-Gal medium until $OD_{600}$ reaching 0.8. SD-Gal liquid medium, containing paraquat (0.75 mM) or diquat (0.75 mM) or no herbicide, was then added to the cell cultures to obtain a 10× dilution. The cultures were shaken at 30 °C, 250 rpm. The $OD_{600}$ of each culture was measured at 12 h, 24 h, 36 h, 48 h, 70 h, and 82 h using a plate reader, to obtain the $OD_{600}$-growth time relationship.

To characterize the transport capabilities of OsLAT1 or OsLAT5, we determined the kinetic constants $K_m$ and $V_{max}$ on BPyHs uptake. The cultured cells were grown until $OD_{600} = 1.0$ in the SD-Gal liquid medium. Then, the cells were harvested, washed three times with the uptake buffer (0.333 mM MES, pH 5.7, 2% galactose), and resuspended in the same buffer at a cell density of $5 \times 10^7$ cells per 100 µL. Uptake was initiated by adding paraquat or diquat to 2 mL culture to certain concentrations (5 µM, 10 µM, 15 µM, 20 µM, 30 µM, and 50 µM). Cells were shake-cultured for 1 h at 250 rpm and then centrifuged to discard supernatant. Subsequently, the cells were washed five times with ice-cold uptake buffer to remove exogenous herbicides and stop uptake. Finally, the cells were homogenized in methanol/water (v/v: 1/1), evaporated in vacuum and resuspended in acetonitrile/water/formic acid (5/14/1) for analysis in ultra-performance liquid chromatography-tandem mass spectrometry (UPLC-MS/MS). Michaelis-Menten parameters were obtained by a non-linear regression method.

## Plant materials

The study used two rice cultivars, Zhonghua11 and Guiyu NO.11, and the mutants and overexpression plants were generated by EDGENE

Biotechnology (Wuhan) Co., Ltd. *oslat1* mutants (*oslat1-a* and *oslat1-b*), *oslat3* mutants (*oslat3-a* and *oslat3-b*), *oslat5* mutants (*oslat5-a* and *oslat5-b*) and *oslat7* mutants (*oslat7-a* and *oslat7-b*) (Zhonghua11 background), as well as GY-*oslat5* mutant (Guiyu NO.11 background), were generated by CRISPR/Cas9 technology and identified by sequencing PCR products (Supplementary Fig. 11) which were amplified with gene-specific primers (Supplementary Table 1). The pCAMBIA1300-35S vector, carrying different *OsLAT* genes, was transformed into Zhonghua11 with *Agrobacterium tumefaciens*-mediated transformation. The overexpression of rice, harboring each *OsLAT* gene was confirmed by 50 mg/L hygromycin resistance and gene expression level analysis (Supplementary Table 1), two lines with relatively high expression levels (*OsLAT1*-OE-1 and *OsLAT1*-OE-2, *OsLAT3*-OE-1 and *OsLAT3*-OE-2, *OsLAT5*-OE-1 and *OsLAT5*-OE-2, *OsLAT7*-OE-1 and *OsLAT7*-OE-2) were selected for further study.

## BPyHs resistance seed germination assay
Seeds were rinsed with 70% (v/v) ethanol for 3 min, and then with 30% (w/v) sodium hypochlorite for 30 min for surface disinfection. Next, the sterilized seeds were washed several times with sterile water, and plated on MS medium (4.4 g/L M&S basal medium with vitamins, 30 g/L sucrose, and 3.5 g/L phytagel at a final pH of 5.8). To assess the effects of BPyHs on seedling morphology, the seeds were germinated on MS medium containing paraquat or diquat (0.05 and 0.10 µM) for 2 weeks. The seedling morphologies were recorded and photographed. The growth conditions were as follows: 14 h light /10 h dark cycle, 25 °C, 300 µmol·m$^{-2}$·s$^{-1}$ light intensity.

## Indoor crop-weed competition
Guiyu NO.11, GY-*oslat5* and weeds (Gramineae, Cyperaceae and broadleaf weeds) were sowed in pots at the same time. The soil was treated with 1‰ CPAM-BPyHs, of which the concentrations of paraquat or diquat were set to 0.5, 1.0, 2.5, 5.0 mM. The height of rice plant and number of weeds were recorded after 7 days. All the plants were grown under greenhouse conditions: 14 h light/10 h dark cycle, 30 °C in light/28 °C in dark, light intensity > 350 µmol·m$^{-2}$·s$^{-1}$.

## Field crop-weed competition
Rice seeds were sowed in fields (Conghua, Guangzhou, Guangdong Province, China) and 1‰ CPAM-diquat were subsequently sprayed onto the soil surface. The diquat dose was 2.5 and 5.0 mM. The field management followed normal DSR agronomic practices. On the 10th or 30th day after application, the rice growth and weed control were investigated.

## Field trial
The trial site was set in Niujiaolun village, Yiyang, Hunan Province, China, where rice was planted all year round. We selected two adjoining fields, one served as control group with an area of 667 m$^2$, another served as experimental group with an area of 800 m$^2$. Fields preparation included irrigation to bring the soil moisture content to 70–80% (no water layer) and leveling the paddy field. GY-*oslat5* seeds were soaked with water for 12 h and then germinated at 28 °C for 24 h in darkness. A manual rice direct seeding machine was used for sowing, with a row spacing of 28.5 cm and a seed amount of 50 kg/hm$^2$. After that, diquat was dissolved in ‰ CPAM solution to obtain a dose of 3800 g a.i./hm$^2$. A knapsack electric sprayer with high-pressure pump (3WBD-20L, CHNONLI) was employed to evenly spray CPAM-diquat onto the soil surface, while water was sprayed as control. The fields were irrigated after two weeks and field management followed normal DSR agronomic practices. On the 23rd day, the herbicidal activity of CPAM-diquat was evaluated. Six sites (1 m$^2$) were randomly selected in each field to survey the number of Gramineae, Cyperaceae and broadleaf weeds separately.

## Reporting summary
Further information on research design is available in the Nature Portfolio Reporting Summary linked to this article.

## Data availability
Data supporting the findings of this work are available within the paper and its Supplementary Information files. A reporting summary for this Article is available as a Supplementary Information file. Source data are provided with this paper.

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

## Acknowledgements

We thank Xingshan Tian, Xiuying He, Zhanhua Lu (Guangdong Academy of Agricultural Sciences) for help with assessment of herbicide resistance and rice cultivation in field; Hong Tang (Yiyang Academy of Agricultural Sciences) for help with field trial in Hunan. F.L. was funded by the National Key Research and Development Program of China (Grant No. 2023YFD1401100), F.L. and R.C. were funded by 2024 Guangdong Rural Revitalization Strategy Special Fund Seed Industry Project (First Batch) (Yue Cai Nong [2024] No. 83), F. L. was funded by the National Natural Science Foundation of China (Grant No. 32072451), H.X., F.L. and L.Z. were funded by Research Fund of State Key Laboratory of Green Pesticide (GPLSCAU2024003).

## Author contributions

H.X., F.L. and L.Z. designed the study. H.X. was responsible for project administration and supervision. R.C. and C.L. evaluated the herbicidal activities indoor and in the field, performed soil adsorption assay of CPAM-BPyHs. R.C. and D.Z. conducted yeast assay and characterized the herbicide resistance of transgenic plants. L.Z. and C.L. tested supramolecular materials and performed MD simulations. R.C., D.Z. and G.Y. constructed BPyHs-resistant rice. R.C. wrote the original draft. H.X., F.L. and L.Z. revised the draft.

## Competing interests

Patent application related to this work has been submitted (Patent applicant institution: South China Agricultural University; Name of inventors: Hanhong Xu, Lingda Zeng, Chaozheng Li, Fei Lin, Ronghua Chen, Yinghao Zhang; Application NO. 2023114526878). The patent protects a method for preventing the inactivation of bipyridyl herbicides in soil by combining with cationic polymer materials including CPAM. The remaining authors declare no competing interests.
