## [Peer Review file · Nature Communications]

Supramolecular Pre-Emergence Herbicide: Towards Efficient and Persistent Weed Control

Corresponding Author: Professor Lingda Zeng

Version 0:

Reviewer comments:

Reviewer #1

(Remarks to the Author)

The manuscript titled "Combining transport-deficient rice and supra-soil-sealer: For weed control in direct-seeded rice" reports the combination of rice OsLAT1 or OsLAT5 knockout mutants with cationic polyacrylamide nanoparticles absorbed with bipyridyl herbicides for weed control in direct-seeded rice. This technical integration holds promise for enhancing weed management in rice cultivation. However, this review highlights several concerns:

1. **Novelty:** While LAT family members, such as AtRMV1 and OsPAR1, have been previously studied for paraquat transport in Arabidopsis and rice, respectively. Their potential application for paraquat resistance has been demonstrated in rice (OsPUTs). It remains unclear whether OsLAT1 and OsLAT5 have been previously reported. The authors should provide clarification on the novelty of their findings. If OsLAT1 and OsLAT5 have not been previously studied, the manuscript should focus on thoroughly characterizing these two members to demonstrate novelty. The current manuscript lacks sufficient experimental support in this regard.
2. **Timing of Herbicide Application:** Bipyridyl herbicides typically target photosynthetic tissues by inducing a burst of reactive oxygen species (ROS) and are traditionally applied to leaves. Therefore, the necessity and cost-effectiveness of soil application of these herbicides should be addressed.
3. **Potential Conflicting Results:** The subcellular localization of OsLAT1 and OsLAT5 in yeast has not been demonstrated. If their localizations resemble those in Arabidopsis protoplasts, the expression of OsLAT5 should not affect paraquat uptake in yeast or rice seedlings. Clarification on this aspect is essential to avoid potential conflicting results.

Addressing these concerns will strengthen the manuscript and contribute to its scientific rigor and impact.

Reviewer #2

(Remarks to the Author)

The abstract, although it is summarizing the points of the manuscript in a brief and concise manner, needs further improvement to better underline the impact that such a study could have. The first and last two sentences are however inappropriate, so the authors should find better formulations for these points.

The first sentence of abstract and Introduction both are exactly the same which is very strange.....?, authors must reframe these sentence's. The Introduction part is way too short!. The authors have described some irrelevant aspects while not putting enough accents in other parts that are more relevant to their context. There are many repetitive statements that can be removed and instead, the authors should focus more on key messages. The purpose of the study should be also better explained.

I would however like to read/follow/evaluate the result section again, after the authors make completely sure of what they are presenting, in a proper language. I also suggest the authors to re-think the way (graphical, section division, total number of figures and tables in the manuscript) they are presenting their results.

Discussion section is very short, author's need to discuss the results properly to make a better connection of the hypothesis. The discussions are often repetitive with what was also said in the introduction. Many parts do not have anything to do with the data presented, being just an enumeration of other experiments. I am highly suggesting a focused discussion, underling

the main points of interest. Authors un-necessarily stretched the discussion section but could not justify their findings. I will ponder over this decision once I can see a manuscript properly made, with fitted language (both scientifically and grammarly), figures, presentation, and solid discussion.

Conclusion section must be included

The treatments especially should be presented in a better manner

What is the basis of dose selection of treatment in this study...?

Why OsLAT1 and OsLAT5 gene...?

The language needs substantial improvements. Articles (the, a, -s) are either missing or misplaced. Many sentences have a strange wording that makes the manuscript difficult to read

Reviewer #3

(Remarks to the Author)

This is an interesting piece of work intending to provide an additional and novel tool for managing weeds, particularly for direct seeded rice in which these species undoubtedly constitute the most important production constraint. The molecular work has been conceived and executed having in mind this major agronomic problem, with the possibility of bringing the outcome of basic research into the hands of rice farmers. The selection of bipyridylum herbicides is unfortunate. They are under serious scrutiny worldwide because of their toxicity to applicators and perceived ecological damage. Indeed, paraquat is a restricted use pesticide in most parts of the world and has been banned in several countries already; many are considering removing it from the market. I understand that China banned the domestic use of paraquat, although they continue producing it for the international market. No transgenic or edited herbicide-resistant rice is grown anywhere in the world and there will be major hurdles to register the proposed rice materials. Therefore, the technology reported in this paper will not be adopted.

There are several conceptual problems in the introduction when justifying the work reported here.

L39-41. Reference 14 deals with the effect of the level of organic matter in the soil on the efficacy of two pre-emergence herbicides (flufenacet and pendimethalin) in controlling *Alopecurus myosuroides*. Nowhere in the paper by Metcalfe et al. (ref 14) the authors refer to a soil sealer or imply the existence of something like it. Therefore, attributing such definition to them can be considered as a misrepresentation. The concept of soil sealer, in fact, does not exist in the herbicide literature. Herbicides that remain active (i.e. they keep controlling weeds for some time) in the soil are called residual herbicides. They are said to have the property of being persistent (i.e. they have long-lasting activity). Herbicides that tightly bind to soil colloids, such as paraquat, have no soil activity (in relation to weed control). The term "(supra) soil sealer" is completely inappropriate.

L42-43. There are plenty of preemergence herbicides registered for use in rice worldwide; they belong to several chemical and mode-of-action groups, being quite useful for preventing and managing the evolution of herbicide resistance. The sentence does not clarify if germination refers to the crop or to the weeds. Indeed, preemergence herbicides are often sprayed (usually in combination with post-emergence ones) after the crop has emerged to control late germinating weeds. This requires careful timing and good knowledge about the soil seed bank dynamics and the biology of the weeds. Farmers often through experience can decide on the appropriate timing for such treatments.

L43-44. Reducing the diversity of herbicides used to manage weeds is a recipe for resistance evolution. Over-reliance on a single product (or group of chemicals sharing the same mode of action or metabolic degradation) is highly conducive to resistance evolution. There are already 76 cases of resistance evolution to BPyHs among 33 species of weeds (see www.weedscience.org).

L111 Here and elsewhere, authors refer to the effect of the herbicides "during germination." Bipyridylum herbicides require a functional Photosystem I (PSI) to cause phytotoxicity. PSI develops after germination, at very early stages of growth. It is the young seedling that can be affected by these compounds that promote the formation of toxic reactive oxygen species.

L112-114. There is a problem with the basic nomenclature of rice aerial organs. When authors refer to "leaf" they probably mean the "leaf blade" (the other part of the leaf is the sheath); the sheath encircles the culm ("stem").

L114-116. Long distance translocation of systemic herbicides occurs through the apoplast and symplast, of which the vascular systems of the plant are major components. Movement among organs is mostly dependent on the physical-chemical characteristics of the herbicide. Root absorbed photosynthetic inhibitors (such as triazines and substituted ureas) move mostly in the apoplast but they should enter the symplast to carry out their biological activity (inhibition of photosynthesis). Paraquat and diquat are applied as postemergence herbicides, to the foliage of plants whose control is intended. In the presence of light, which is required for their action, they only move short distances. When applied at night or in the dark they can enter the apoplast and distribute more freely (for example from a treated leaf to an adjacent untreated leaf). Given these attributes, I expect the authors will provide a clear explanation on how the overexpression of OsLAT1 enhances both root uptake and the root-to-shoot transport of diquat and paraquat.

L128-129. Authors indicate that they determined the overall fitness and yield of conventional GY rice and GY-oslat5 rice "under normal field conditions" referring the reader to fig. S6A. First of all, fitness must be measured in competition using an appropriate plant density (see review by Vila-Aiub 2019). The measurements taken at most only give an indication of productivity. Also, fig. S6A shows plants growing in pots (unclear if as single individuals or not), which does not correspond at all with a "normal field condition."

L149-151 Please refer to my previous comments regarding soil behaviour and weed management concepts.

L154-162 I am not fully convinced of the pre-emergence activity of the CPAM-BPyHs combination. There is no indication (in M&M or in this section) of a CPAM-alone control. Does CPAM at the 1% (I assume by volume) dose have any herbicidal activity? In L383 authors indicate that they recorded toxicity symptoms at 7 days after application (DAA). Plant height is not a good measurement of BPyHs phytotoxicity and etiolation is not a distinct symptom caused by these herbicides. Rapid tissue necrosis is the typical symptom that develops soon after paraquat/diquat spraying. Under conditions that allow some level of translocation (see previous comment), foliage necrosis occurs after exposure to sunlight. Surviving plants shown in Fig S7

look healthy despite any growth reduction. Authors mention control of other species (e.g. *Cyperus rotundus*, *Eclipta prostrata* and *Echinochloa colona*). I see a broadleaf species in the photographs of Fig S7 that is not mentioned or identified in the caption, not showing apparent damage (although they are not present at higher doses of the herbicides in mixture with CPAM). No data at all is provided on the mentioned weeds. I would not expect being able to gather data about *C. rotundus* at 7 days. Seed of this species is by far sterile and if planted by tubers, time required for their sprouting is much longer than a week. I am also surprised that *E. colona* was tested instead of *E. crus-galli*. *E. colona* is seldom found in rice fields in China, whereas the diverse botanical varieties of *E. crus-galli* are widespread. Rather than etiolation, from the photograph I am more interested in the effect of the treatments on the density of the rice plants. At 7 DAA, if emergence was affected by the treatments, a proportion of the emerging seedlings would be dead, which is the outcome a farmer would expect.

L163 What do you mean by “natural” condition?

L163-173. Were the evaluated weeds of the three major groups (cyperaceae, POACEAE, not gramineae, and broadleaf) planted or volunteers? Do you have any data about the densities of these weeds in each of the two emergence flushes? Field “sustainability” of the method deserves thorough discussion.

L180-189 There is no indication of the origin of the soil used (type, texture, organic matter content, cation exchange capacity) and the moisture level at the time of application. These characteristics are important in relation to adsorption of herbicides to soils. Vast amounts of paraquat are required to saturate the soil binding capacity (see Moyer and Lindwall 1985. *Can. J. Soil Sci.* 65: 523-529), therefore it is unlikely that biologically active (herbicidal) concentrations would be available for root uptake under field conditions. Distilled-water extraction would reflect the traces of paraquat/diquat available for root uptake; determination of soil residues usually requires harsh (sulfuric acid) extraction. The assertion that CPAM decreases herbicide adsorption ensuring biological soil activity of BPyHs is speculative. It must be proven. Perhaps a dose response curve with water extracts using an indicator plant such as *Lemna* can be used to prove it. In L182 the dose of the herbicides applied should be given to relate to the residue found. Since authors are following a fully agronomic approach to stress the value of their work, dose must be given in g of active per hectare. Indicating just a concentration of the spraying preparation is not appropriate, particularly since the authors do not provide any indication of the total volume used per unit area.

L201-202. Is there any adsorption curve to verify saturation?

Discussion section

Discussion is completely absent. The first paragraph is a very general, inconsequential literature review. The second one is a sales pitch lacking any substance. I think that my comments throughout the manuscript can provide the authors enough food for a meaningful discussion. Additionally, it is important that the authors consider key issues such as the possibility of gene flow from the improved varieties to weedy rice, the most aggravating weed problem emerging with direct seeding. Have they considered any possibility of molecular mitigation of gene flow? It would be also interesting to know how the authors perceive the resistance mechanism conferred through editing in relation to the widespread natural selection of paraquat resistant weeds that are able to sequester the herbicide away from their active site. Finally, in explaining long-term soil activity, the reader most likely would expect some discussion about the actual soil behaviour of the herbicides. To be effective over a month or so, there ought to be an equilibrium between bound paraquat/diquat and that free in solution available for uptake. The soil is a complex matrix with herbicides interacting with all types of colloids, including expanding lattice 2:1 clays from which the herbicide cannot be retrieved.

A few additional minor points:

L291. What was the procedure for stopping paraquat/diquat uptake?

Caption of Fig S2 A is unclear. What do you mean by “collected”?

In the caption of figure 1, there is no indication of what “a - d” and “2” “8” “9” and “10” mean. Kindly, clarify this in the caption.

The same applies for S3 (in relation to “e, f, g, and h” and “4, 6, 7, and 8”).

Caption of figure S4. It is important to indicate that the uptake studies were conducted in hydroponic conditions, herbicides being absorbed through the roots.

Caption of figure S6B. Please indicate what the three groups of seed are. I assume they correspond to the mature (raw) seed, dehulled seed and polished grain for consumption.

Caption of figure S7. Please clearly state how the herbicides and adjuvant were applied (i.e. resembling a pre-emergence application). Was rice planted before spraying?

Version 1:

Reviewer comments:

Reviewer #1

(Remarks to the Author)

Regarding the authors' responses:

OsPAR1 and OsPUTs were all previously reported OsLATs.

The authors compared the effectiveness of CAMP-HPyHs with foliar spray (Figure 5); however, the presentation of this data appears selectively curated, minimizing the demonstrated efficacy of foliar-applied HPyHs by omitting visual evidence of their weed-killing effects. Figure 5F lacks a legend.

Reviewer #2

(Remarks to the Author)

Authors have included all the suggestions and have made significant improvement during the revision of MS

Reviewer #3

(Remarks to the Author)

The new version of the paper is completely renovated, and the work is justified with an entirely different approach. The first version emphasized some major points: (1) The relevance of rice as a major crop and the limitations imposed by weeds when grown under direct seeding, (2) The development of a transgenic rice resistant to bipyridylum herbicides by knocking out the gene responsible for its intracellular transport, and (3) The development of a system allowing for residual activity of bipyridylum herbicides using cationic polyacrylamide. Integration of both experimental technologies was proposed as a major achievement for the control of weeds in rice and the prevention of herbicide resistance evolution.

Although changes in the focus of the paper seem to accommodate objections from referees to the first version, including (post fact) adjusting of objectives, the new manuscript better describes and emphasizes the agronomic value of the research: making a highly soil inactive herbicide group (bipyridyliums) gain some level of preemergence activity and persistence. Key to modifying the soil behavior of bipyridylum herbicides is weakening inactivation by competitive binding and charge modulation.

I have made comments and suggestions to the manuscript in the annotated version attached. In addition to what is mentioned there, I would like to stress once again my reservations regarding terminology, particularly that related to the "soil sealing." Further to my objections regarding the term in relation to the soil behaviour of herbicides (preemergence activity and residuality/persistence), I also would like to add the perspective from soil science. Soil surface sealing is recognized as detrimental to seed germination. A thin seal develops when soil particles are dispersed mostly by rain clogging pore spaces in the soil matrix; it also occurs when soil particles detach and are deposited down with water runoff (see 10.1097/00010694-195804000-00002 and 10.2136/sssaj1981.03615995004500050004x). Interestingly, PAMs are used to prevent the formation of these soil seals.

It would be valuable to include in the paper experimental data regarding the adsorption of CPAM on soils (isotherms). Of interest is also to know how many positive charges are found in the CPAM polymer used in this work (charge density) and to provide detailed information about the methods to spray the polymer as it is established that shearing of PAM molecules can happen, reducing its viscosity and effectiveness.

Description of methods should be improved. Methods are supposed to provide enough information as for a person knowledgeable in the state of the art to repeat the experiment. Most descriptions of experimental procedures are not enough to achieve this purpose. Often, there is inconsistency in the interpretation and presentation of results and observations. For example, in L97-98 authors state that contents of free paraquat or diquat in soil decreased over time referring the reader to Fig. 3C-D. The figure, however, refers to paraquat and diquat degradation. Do they mean microbial or chemical degradation? Also, the figure caption does not indicate what the insert graphs represent and how they compare to the main ones. The discussion is limited; authors should make sure that all relevant results are addressed. Emphasis is placed on the positive effect of competitive binding and charge modulation for providing preemergence activity of pesticides, in general, against weeds (and pathogens). The extrapolation to pesticides as a group is not based on experimental evidence. The work reported here only included the bipyridylum herbicides (that are an exception for being positively charged molecules with a planar conformation) and a few weeds. Discussion should focus on the actual experimental data. There is no mention of possible negative environmental effects of using cationic PAM, which is known for its toxicity to fish.

Finally, I should indicate that I see no improvement in the English quality in this version of the paper. Once again, I strongly suggest having a proficient English speaker or a specialized company thoroughly revising it.

Version 2:

Reviewer comments:

Reviewer #1

(Remarks to the Author)

The authors have adequately addressed my concerns. It would be better if the biosafety issue of CPAM is clarified and the input cost is contrasted between CPAM-based method and foliar spray. Reference 46 (Chen et al., 2025) seems irrelevant.

Reviewer #3

(Remarks to the Author)

The authors have fully considered my suggestions substantially improving the manuscript. Language has also been improved. I have no further observations to make on the paper.

Response to the reviewer's comments

Reviewer #1 (Remarks to the Author):

The manuscript titled "Combining transport-deficient rice and supra-soil-sealer: For weed control in direct-seeded rice" reports the combination of rice OsLAT1 or OsLAT5 knockout mutants with cationic polyacrylamide nanoparticles absorbed with bipyridyl herbicides for weed control in direct-seeded rice. This technical integration holds promise for enhancing weed management in rice cultivation.

1. **Novelty:** While LAT family members, such as AtRMV1 and OsPAR1, have been previously studied for paraquat transport in Arabidopsis and rice, respectively. Their potential application for paraquat resistance has been demonstrated in rice (OsPUTs). It remains unclear whether OsLAT1 and OsLAT5 have been previously reported. The authors should provide clarification on the novelty of their findings. If OsLAT1 and OsLAT5 have not been previously studied, the manuscript should focus on thoroughly characterizing these two members to demonstrate novelty. The current manuscript lacks sufficient experimental support in this regard.

Response: We appreciate your positive comments and careful review. As the novelty of our work mainly lies in the fabrication of adsorbed-but-active supra-soil-sealers, we have rewritten the article. The gene functions of OsLAT1 and OsLAT5 are not central to the new article and have not been deeply characterized. To the best of our knowledge, it is the first time the ability of OsLATs family in transporting bipyridyl herbicides has been compared, which is valuable for developing bipyridyl herbicides-resistant rice cultivars through precise editing of OsLATs.

2. **Timing of Herbicide Application:** Bipyridyl herbicides typically target photosynthetic tissues by inducing a burst of reactive oxygen species (ROS) and are traditionally applied to leaves. Therefore, the necessity and cost-effectiveness of soil application of these herbicides should be addressed.

Response: As suggested, a comparison between soil sealing and foliar spraying has been conducted in our new article. Soil sealing with CPAM-BPyHs offered practical advantages over foliar spraying with BPyHs.

3. **Potential Conflicting Results:** The subcellular localization of OsLAT1 and OsLAT5 in yeast has not been demonstrated. If their localizations resemble those in Arabidopsis protoplasts, the expression of OsLAT5 should not affect paraquat uptake in yeast or rice seedlings. Clarification on this aspect is essential to avoid potential conflicting results.

Response: With the change of the theme and content of the new article, the subcellular localization has not been thoroughly researched, but it has been demonstrated that the expression of OsLAT5 affects paraquat uptake in rice seedlings.

Reviewer #2 (Remarks to the Author):

The abstract, although it is summarizing the points of the manuscript in a brief and concise manner, needs further improvement to better underline the impact that such a study could have. The first and last two sentences are however inappropriate, so the authors should find better formulations for these points.

The first sentence of abstract and Introduction both are exactly the same which is very strange.....?, authors must reframe these sentence's. The Introduction part is way too short! The authors have described some irrelevant aspects while not putting enough accents in other parts that are more relevant to their context. There are many repetitive statements that can be removed and instead, the authors should focus more on key messages. The purpose of the study should be also better explained.

I would however like to read/follow/evaluate the result section again, after the authors make completely sure of what they are presenting, in a proper language. I also suggest the authors to re-think the way (graphical, section division, total number of figures and tables in the manuscript) they are presenting their results.

Discussion section is very short, author's need to discuss the results properly to make a better connection of the hypothesis. The discussions are often repetitive with what was also said in the introduction. Many parts do not have anything to do with the data presented, being just an enumeration of other experiments. I am highly suggesting a focused discussion, underling the main points of interest. Authors un-necessarily stretched the discussion section but could not justify their findings. I will ponder over this decision once I can see a manuscript properly made, with fitted language (both scientifically and grammarly), figures, presentation, and solid discussion.

Conclusion section must be included.

The treatments especially should be presented in a better manner.

Response: Thanks for your comments and careful review. We have rewritten the article and improved upon the issues.

What is the basis of dose selection of treatment in this study...?

Response: The relationship between herbicidal activity and CPAM concentrations was added in Fig. 1 to give an appropriate dose.

Why OsLAT1 and OsLAT5 gene...?

Response: The reason for gene selection has been explained in the new article.

The language needs substantial improvements. Articles (the, a, -s) are either missing or misplaced.

Many sentences have a strange wording that makes the manuscript difficult to read.

Response: As suggested, the language has been polished through.

Reviewer #3 (Remarks to the Author):

This is an interesting piece of work intending to provide an additional and novel tool for managing weeds, particularly for direct seeded rice in which these species undoubtedly constitute the most important production constraint. The molecular work has been conceived and executed having in mind this major agronomic problem, with the possibility of bringing the outcome of basic research into the hands of rice farmers. The selection of bipyridylium herbicides is unfortunate. They are under serious scrutiny worldwide because of their toxicity to applicators and perceived ecological damage. Indeed, paraquat is a restricted use pesticide in most parts of the world and has been banned in several countries already; many are considering removing it from the market. I understand that China banned the domestic use of paraquat, although they continue producing it for the international market. No transgenic or edited herbicide-resistant rice is grown anywhere in the world and there will be major hurdles to register the proposed rice materials. Therefore, the technology reported in this paper will not be adopted.

Response: Thanks very much for the positive evaluation of the significance of our research. The article has been rewritten to not only provide methods for weed management but also propose strategies for designing pesticides against soil inactivation and drifting with water.

In the selection of herbicides, bipyridyl herbicides (BPyHs) continue to hold a significant market share globally, with the advantages of being highly effective, broad-spectrum and residue-free. In China, diquat, a member of the BPyHs, is widely accepted by farmers. Additionally, many new methods have been developed to address the toxicity issues of BPyHs (*Nat. Commun.*, **2018**, *9*, 2967; *J. Am. Chem. Soc.*, **2012**, *134*, 19489). Therefore, we believe this research possesses practical application value, and our strategy for fabricating adsorbed-but-active pesticide is innovative and scientifically significant.

In the new article, direct-seeded rice serves merely as an example to illustrate the effects of CPAM-BPyHs. Therefore, we rapidly constructed a BPyHs-resistant rice cultivar with gene editing, and similar methods can be applied to other crops. As for the feasibility of gene editing, on the one hand, gene editing is not the only way to obtain resistant crops, genetic screening and hybridization can also achieve herbicide resistance. On the other hand, gene-edited plants are gradually being embraced worldwide, including in China. In 2023, the Ministry of Agriculture and Rural Affairs issued China's first safety certificate for plant gene editing. We have strong reason to believe that our work holds significant application value.

There are several conceptual problems in the introduction when justifying the work reported here.

L39-41. Reference 14 deals with the effect of the level of organic matter in the soil on the efficacy of two pre-emergence herbicides (flufenacet and pendimethalin) in controlling *Alopecurus myosuroides*. Nowhere in the paper by Metcalfe et al. (ref 14) the authors refer to a soil sealer or imply the existence of something like it. Therefore, attributing such definition to them can be considered as a misrepresentation. The concept of soil sealer, in fact, does not exist in the herbicide literature. Herbicides that remain active (i.e. they keep controlling weeds for some time) in the soil are called residual herbicides. They are said to have the property of being persistent (i.e. they have long-lasting activity). Herbicides that tightly bind to soil colloids, such as paraquat, have no soil activity (in relation to weed control). The term “(supra) soil sealer” is completely inappropriate.

Response: In our article, we borrow the concept of soil sealing from environmental science and use the term “(supra) soil sealer” to emphasize that herbicides can continuously control weeds before their appearance without drift with water, just like sealing the soil. We believe that “soil-sealer” better captures the weed control characteristics of CPAM-BPyHs and have defined it in the text.

L42-43. There are plenty of preemergence herbicides registered for use in rice worldwide; they belong to several chemical and mode-of-action groups, being quite useful for preventing and managing the evolution of herbicide resistance. The sentence does not clarify if germination refers to the crop or to the weeds. Indeed, preemergence herbicides are often sprayed (usually in combination with post-emergence ones) after the crop has emerged to control late germinating weeds. This requires careful timing and good knowledge about the soil seed bank dynamics and the biology of the weeds. Farmers often through experience can decide on the appropriate timing for such treatments.

Response: We have rewritten the article and improved the expression.

L43-44. Reducing the diversity of herbicides used to manage weeds is a recipe for resistance evolution. Over-reliance on a single product (or group of chemicals sharing the same mode of action or metabolic degradation) is highly conducive to resistance evolution. There are already 76 cases of resistance evolution to BPyHs among 33 species of weeds (see www.weedscience.org).

Response: The resistance of weeds is related to many factors. Generally, controlling weeds when they are freshly germinated can delay the development of resistance. Moreover, our method can also be used in conjunction with other methods to delay the development of resistance.

L111 Here and elsewhere, authors refer to the effect of the herbicides “during germination.” Bipyridylium herbicides require a functional Photosystem I (PSI) to cause phytotoxicity. PSI develops after germination, at very early stages of growth. It is the young seedling that can be affected by these compounds that promote the formation of toxic reactive oxygen species.

Response: We have rewritten the article and improved the expression.

L112-114. There is a problem with the basic nomenclature of rice aerial organs. When authors refer to “leaf” they probably mean the “leaf blade” (the other part of the leaf is the sheath); the sheath encircles the culm (“stem”).

Response: As suggested, we have improved the expression.

L114-116. Long distance translocation of systemic herbicides occurs through the apoplast and symplast, of which the vascular systems of the plant are major components. Movement among organs is mostly dependent on the physical-chemical characteristics of the herbicide. Root absorbed photosynthetic inhibitors (such as triazines and substituted ureas) move mostly in the apoplast but they should enter the symplast to carry out their biological activity (inhibition of photosynthesis). Paraquat and diquat are applied as postemergence herbicides, to the foliage of

plants whose control is intended. In the presence of light, which is required for their action, they only move short distances. When applied at night or in the dark they can enter the apoplast and distribute more freely (for example from a treated leaf to an adjacent untreated leaf). Given these attributes, I expect the authors will provide a clear explanation on how the overexpression of OsLAT1 enhances both root uptake and the root-to-shoot transport of diquat and paraquat.

Response: We have rewritten the article to focus on providing a strategy to fabricate adsorbed-but-active supra-soil-sealers. We are so sorry that we cannot provide a clear explanation about this question due to the limitations of the article's theme and length.

L128-129. Authors indicate that they determined the overall fitness and yield of conventional GY rice and GY-oslat5 rice “under normal field conditions” referring the reader to fig. S6A. First of all, fitness must be measured in competition using an appropriate plant density (see review by Vila-Aiub 2019). The measurements taken at most only give an indication of productivity. Also, fig. S6A shows plants growing in pots (unclear if as single individuals or not), which does not correspond at all with a “normal field condition.”

Response: We have rewritten the article, in which direct-directed rice is just an example to show the herbicidal effect of CPAM-BPyHs. The evaluation of agronomic traits is no longer involved.

L149-151 Please refer to my previous comments regarding soil behavior and weed management concepts.

Response: We have rewritten the article and improved the expression.

L154-162 I am not fully convinced of the pre-emergence activity of the CPAM-BPyHs combination. There is no indication (in M&M or in this section) of a CPAM-alone control. Does CPAM at the 1% (I assume by volume) dose have any herbicidal activity? In L383 authors indicate that they recorded toxicity symptoms at 7 days after application (DAA). Plant height is not a good measurement of BPyHs phytotoxicity and etiolation is not a distinct symptom caused by these herbicides. Rapid tissue necrosis is the typical symptom that develops soon after paraquat/diquat spraying. Under conditions that allow some level of translocation (see previous comment), foliage necrosis occurs after exposure to sunlight. Surviving plants shown in Fig S7 look healthy despite any growth reduction. Authors mention control of other species (e.g. *Cyperus rotundus*, *Eclipta prostrata* and *Echinochloa colona*). I see a broadleaf species in the photographs of Fig S7 that is not mentioned or identified in the caption, not showing apparent damage (although they are not present at higher doses of the herbicides in mixture with CPAM). No data at all is provided on the mentioned weeds. I would not expect being able to gather data about *C. rotundus* at 7 days. Seed of this species is by far sterile and if planted by tubers, time required for their sprouting is much longer than a week. I am also surprised that *E. colona* was tested instead of *E. crusgalli* is seldom found in rice fields in China, whereas the diverse botanical varieties of *E. crusgalli* are widespread. Rather than etiolation, from the photograph I am more interested in the effect of the treatments on the density of the rice plants. At 7 DAA, if emergence was affected by the treatments, a proportion of the emerging seedlings would be dead, which is the outcome a farmer would expect.

Response: CPAM has no significant influence on weed growth, as shown in Figure. 1.

Our goal is to control weeds before their appearance, so the germination rate of weeds was used to assess the weed control effectiveness of CPAM-BPyHs rather than the level of foliage necrosis. Efficient and persistent (30 days) weed control of CPAM-BPyHs was verified several times in pots, small fields and large fields.

Plant height was only used to assess rice resistance rather than BPyHs phytotoxicity, while the expression " etiolation" has been removed.

The weeds for the indoor experiments were grown from seeds collected from the fields, while the weeds in the field trials grew naturally. We refined our statement to focus on the three main types of weeds (Gramineae, Cyperaceae, and broadleaf weeds) rather than specific genera, to avoid controversy.

We did not observe a significant effect of the treatment on the density of resistant rice plants.

L163 What do you mean by “natural” condition?

Response: Natural conditions mean that there is no additional human intervention other than herbicide application and rice cultivation.

L163-173. Were the evaluated weeds of the three major groups (Cyperaceae, POACEAE, not Gramineae, and broadleaf) planted or volunteers? Do you have any data about the densities of these weeds in each of the two emergence flushes? Field “sustainability” of the method deserves thorough discussion.

Response: Weeds grow naturally in the field. The expression "the two emergence flushes" has been removed to make the article clearer. Field “sustainability” of the method has been analyzed in more detail as suggested.

L180-189 There is no indication of the origin of the soil used (type, texture, organic matter content, cation exchange capacity) and the moisture level at the time of application. These characteristics are important in relation to adsorption of herbicides to soils. Vast amounts of paraquat are required to saturate the soil binding capacity (see Moyer and Lindwall 1985. Can. J. Soil Sci. 65: 523-529), therefore it is unlikely that biologically active (herbicidal) concentrations would be available for root uptake under field conditions. Distilled-water extraction would reflect the traces of paraquat/diquat available for root uptake; determination of soil residues usually requires harsh (sulfuric acid) extraction. The assertion that CPAM decreases herbicide adsorption ensuring biological soil activity of BPyHs is speculative. It must be proven. Perhaps a dose response curve with water extracts using an indicator plant such as Lemna can be used to prove it. In L182 the dose of the herbicides applied should be given to relate to the residue found. Since authors are following a fully agronomic approach to stress the value of their work, dose must be given in g of active per hectare. Indicating just a concentration of the spraying preparation is not appropriate, particularly since the authors do not provide any indication of the total volume used per unit area.

Response: As suggested, systematic experiments have been performed to figure out the protective mechanism of CPAM against soil inactivation of BPyHs. The new results indicate that CPAM cannot saturate the binding capacity of soil particles but can weaken the soil adsorption strength of soil to BPyHs. The dosage of CPAM-BPyHs has been noted in the caption of Figure. 6.

L201-202. Is there any adsorption curve to verify saturation?

Response: In MD simulations, excess methyl viologen was added to ensure the soil particles reaching saturated adsorption, which makes it feasible to compare the interaction between CPAM, methyl viologen and soil particles.

Discussion section

Discussion is completely absent. The first paragraph is a very general, inconsequential literature review. The second one is a sales pitch lacking any substance. I think that my comments throughout the manuscript can provide the authors enough food for a meaningful discussion. Additionally, it is important that the authors consider key issues such as the possibility of gene flow from the improved varieties to weedy rice, the most aggravating weed problem emerging with direct seeding. Have they considered any possibility of molecular mitigation of gene flow? It would be also interesting to know how the authors perceive the resistance mechanism conferred through editing in relation to the widespread natural selection of paraquat resistant weeds that are able to sequester the herbicide away from their active site. Finally, in explaining long-term soil activity, the reader most likely would expect some discussion about the actual soil behaviour of the herbicides. To be effective over a month or so, there ought to be an equilibrium between bound paraquat/diquat and that free in solution available for uptake. The soil is a complex matrix with herbicides interacting with all types of colloids, including expanding lattice 2:1 clays from which the herbicide cannot be retrieved.

Response: As suggested, we have rewritten the article and elaborated on the strategy for fabricating adsorbed-but-active pesticide, the key highlight of the new article, in the discussion section.

A few additional minor points:

L291. What was the procedure for stopping paraquat/diquat uptake?

Caption of Fig S2 A is unclear. What do you mean by “collected”?

In the caption of figure 1, there is no indication of what “a - d” and “2” “8” “9” and “10” mean. Kindly, clarify this in the caption. The same applies for S3 (in relation to “e, f, g, and h” and “4, 6, 7, and 8”).

Caption of figure S4. It is important to indicate that the uptake studies were conducted in hydroponic conditions, herbicides being absorbed through the roots.

Caption of figure S6B. Please indicate what the three groups of seed are. I assume they correspond to the mature (raw) seed, dehulled seed and polished grain for consumption.

Caption of figure S7. Please clearly state how the herbicides and adjuvant were applied (i.e. resembling a pre-emergence application). Was rice planted before spraying?

Response: As suggested, we have improved the expression.

Response to the reviewer's comments

Reviewer #1 (Remarks to the Author):

1. OsPAR1 and OsPUTs were all previously reported OsLATs.

Response: We appreciate your comments and careful review. OsPAR1 and OsPUTs were reported and previously and related literatures have been cited in the article. Building on their work, we further compared the ability of OsLATs family members to confer resistance to BPyHs in rice and constructed a BPyHs-resistant rice mutant, which have not been reported yet. Although these data are not the central focus of the article, it is meaningful for validating the application potential of combining supramolecular pre-emergence herbicides CPAM-BPyHs with resistant crops in farming. Therefore, we have included these data in our article.

2. The authors compared the effectiveness of CAMP-BPyHs with foliar spray (Figure 5); however, the presentation of this data appears selectively curated, minimizing the demonstrated efficacy of foliar-applied BPyHs by omitting visual evidence of their weed-killing effects. Figure 5F lacks a legend.

Response: Due to the field of view in the images, the effect of foliar spraying appears to be less noticeable. Therefore, we explained in the text that treated weeds quickly wilted and their growth is inhibited (L174-L175). However, soil treatment with CPAM-BPyHs offers practical advantages over foliar spraying with BPyHs.

As suggested, the legend in Figure 5F is added.

Reviewer #2 (Remarks to the Author):

Authors have included all the suggestions and have made significant improvement during the revision of MS

Response: Thanks very much for the positive comments.

Reviewer #3 (Remarks to the Author):

The new version of the paper is completely renovated, and the work is justified with an entirely different approach. The first version emphasized some major points: (1) The relevance of rice as a major crop and the limitations imposed by weeds when grown under direct seeding, (2) The development of a transgenic rice resistant to bipyridylium herbicides by knocking out the gene responsible for its intracellular transport, and (3) The development of a system allowing for residual activity of bipyridylium herbicides using cationic polyacrylamide. Integration of both experimental technologies was proposed as a major achievement for the control of weeds in rice and the prevention of herbicide resistance evolution.

Although changes in the focus of the paper seem to accommodate objections from referees to the first version, including (post fact) adjusting of objectives, the new manuscript better describes and emphasizes the agronomic value of the research: making a highly soil inactive herbicide group (bipyridyliums) gain some level of preemergence activity and persistence. Key to modifying the soil behavior of bipyridylium herbicides is weakening inactivation by competitive binding and charge modulation.

I have made comments and suggestions to the manuscript in the annotated version attached.

Response: We appreciate your positive comments and careful review. We have addressed the comments and suggestions point-by-point in your annotated version.

In addition to what is mentioned there, I would like to stress once again my reservations regarding terminology, particularly that related to the "soil sealing." Further to my objections regarding the term in relation to the soil behaviour of herbicides (preemergence activity and residuality/persistence), I also would like to add the perspective from soil science. Soil surface sealing is recognized as detrimental to seed germination. A thin seal develops when soil particles are dispersed mostly by rain clogging pore spaces in the soil matrix; it also occurs when soil particles detach and are deposited down with water runoff (see 10.1097/00010694-195804000-00002 and 10.2136/sssaj1981.03615995004500050004x). Interestingly, PAMs are used to prevent the formation of these soil seals.

Response: As suggested, we have replaced this term with "pre-emergence herbicide" throughout the manuscript.

It would be valuable to include in the paper experimental data regarding the adsorption of CPAM on soils (isotherms). Of interest is also to know how many positive charges are found in the CPAM polymer used in this work (charge density) and to provide detailed information about the methods to spray the polymer as it is established that shearing of CPAM molecules can happen, reducing its viscosity and effectiveness.

Response: Since the adsorption of CPAM on soils has been reported in the literature, we did not perform experiments on the adsorption of CPAM on soils. Instead, we discussed this aspect in the discussion section and cited the relevant references.

The ionic degree (30%-35%) of CPAM we used was given in the Methods section, which represents the number of cationic repeating units in CPAM.

As suggested, the detailed information about the methods to spray the polymer has been provided in the Methods section.

Description of methods should be improved. Methods are supposed to provide enough information as for a person knowledgeable in the state of the art to repeat the experiment. Most descriptions of experimental procedures are not enough to achieve this purpose. Often, there is inconsistency in the interpretation and presentation of results and observations. For example, in L97-98 authors state that contents of free paraquat or diquat in soil decreased over time referring the reader to Fig. 3C-D. The figure, however, refers to paraquat and diquat degradation. Do they mean microbial or chemical degradation? Also, the figure caption does not indicate what the insert graphs represent and how they compare to the main ones. The discussion is limited; authors should make sure that all relevant results are addressed. Emphasis is placed on the positive effect of competitive binding and charge modulation for providing preemergence activity of pesticides, in general, against weeds (and pathogens). The extrapolation to pesticides as a group is not based on experimental evidence. The work reported here only included the bipyridylium herbicides (that are an exception for being positively charged molecules with a planar conformation) and a few weeds. Discussion should focus on the actual experimental data. There is no mention of possible negative environmental effects of using cationic PAM, which is known for its toxicity to fish.

Response: As suggested, the description of methods has been modified.

Thank you for pointing out the mistake in Fig. 3C-D. In fact, the experiment measured the content of free BPyHs in the soil. We have corrected Fig. 3C-D to avoid any misunderstanding. Additionally, captions for the insert graphs have been added.

As suggested, we have revised the Discussion section to avoid overgeneralization and have included a discussion on the environmental impacts of cationic polyacrylamide (CPAM). Additionally, some results were discussed in the Results section, so they were not reiterated in the Discussion section to avoid redundancy.

Finally, I should indicate that I see no improvement in the English quality in this version of the paper.

Once again, I strongly suggest having a proficient English speaker or a specialized company thoroughly revising it.

Response: As suggested, the language has been polished through by Springer Nature Author Services. (The certificate has been submitted.)

Response to the reviewer's comments

Reviewer #1 (Remarks to the Author):

The authors have adequately addressed my concerns. It would be better if the biosafety issue of CPAM is clarified and the input cost is contrasted between CPAM-based method and foliar spray. Reference 46 (Chen *et al.*, 2025) seems irrelevant.

Response: Thanks very much for your positive comments. The biosafety issue of CPAM have been mentioned in the Discussion section (L280-L282). The input cost comparison between CPAM-based method and foliar spray have been added in the Results section (L189-L191). Reference 46 has been deleted.

Reviewer #3 (Remarks to the Author):

The authors have fully considered my suggestions substantially improving the manuscript.

Language has also been improved. I have no further observations to make on the paper.

Response: Thanks very much for your positive comments.